# Processing of motion boundary orientation in macaque V2

**Heng Ma, Pengcheng Li, Jiaming Hu, Xingya Cai, Qianling Song, Haidong D Lu***

State Key Laboratory of Cognitive Neuroscience and Learning, IDG/McGovern Institute for Brain Research, Beijing Normal University, Beijing, China

**Abstract** Human and nonhuman primates are good at identifying an object based on its motion, a task that is believed to be carried out by the ventral visual pathway. However, the neural mechanisms underlying such ability remains unclear. We trained macaque monkeys to do orientation discrimination for motion boundaries (MBs) and recorded neuronal response in area V2 with microelectrode arrays. We found 10.9% of V2 neurons exhibited robust orientation selectivity to MBs, and their responses correlated with monkeys' orientation-discrimination performances. Furthermore, the responses of V2 direction-selective neurons recorded at the same time showed correlated activity with MB neurons for particular MB stimuli, suggesting that these motion-sensitive neurons made specific functional contributions to MB discrimination tasks. Our findings support the view that V2 plays a critical role in MB analysis and may achieve this through a neural circuit within area V2.

## Introduction

It is an important task to recognize an object when it is in motion. Humans and nonhuman primates have an excellent ability in detection of motion boundaries (MBs) (*Regan, 1986*; *Regan, 1989*). In nonhuman primates, it has been shown that neurons selective for the orientations of MBs are located in the ventral visual pathway (*Marcar et al., 2000*; *Mysore et al., 2006*; *Chen et al., 2016*), but not in the dorsal pathway (*Marcar et al., 1995*). Particularly, it has been shown that area V2 exhibits significant MB responses. Compared with V1, V2 has more MB orientation neurons (*Marcar et al., 2000*) and a significant map for MB orientation (*Chen et al., 2016*). However, other visual areas, including V3 and V4, also have strong responses to MB (*Zeki et al., 2003*; *Mysore et al., 2006*). It is unclear whether all these areas contribute to the eventual perception of MBs. This question is particularly important for V2 since it is the lowest area in the visual processing hierarchy that possesses MB sensitivity.

V2 is the largest extrastriate visual area in primates. It receives inputs from V1 and contains a full retinotopic map (*Gattass et al., 1981*). V2 has different CO stripes (thick, pale, thin), in which neurons exhibit different functional properties and project to different downstream areas (*Shipp and Zeki, 1985*; *DeYoe and Van Essen, 1985*; *Munk et al., 1995*). V2 performs higher-level analysis on visual information in multiple dimensions, for example, shape, color, binocular disparity, and motion. Much of the analysis contributes to figure-ground segregation (e.g., *von der Heydt et al., 1984*; *Zhou et al., 2000*).

In the classical view, visual motion information is processed in the dorsal visual pathway. However, mounting evidences have shown that many other brain areas participate in visual motion processing (*Orban et al., 2003*). In macaque monkeys, there are a large number of direction-selective (DS) neurons in V2. Nevertheless, their functional contributions to visual perception remain unclear. These DS neurons cluster and form functional maps in V2 (*Lu et al., 2010*; *An et al., 2012*). Cooling studies have found that these neurons do not contribute to the motion detection in the dorsal pathway (*Ponce et al., 2008*; *Ponce et al., 2011*). Compared with middle temporal (MT) neurons, V2 DS

*For correspondence:
haidong@bnu.edu.cn

**Competing interests:** The authors declare that no competing interests exist.

neurons have smaller receptive fields (RFs), stronger surround modulation, and higher sensitivity to motion contrast (*Hu et al., 2018*). Thus, these neurons are suitable for figure-ground segregation and/or MB detection (*Chen et al., 2016*; *Hu et al., 2018*). These findings suggest that V2 may calculate the MB orientation using a local DS-to-MB circuit. Testing this hypothesis will also help us to understand the motion processing in V2 and other extra-dorsal areas.

In this study, we trained two monkeys to perform an orientation-discrimination task. From electrode arrays implanted over areas V1 and V2, we recorded neural activity and examined (1) their selectivity to the orientation of MB, (2) their correlation with monkeys' behavioral choice, and (3) neuronal couplings between DS neurons and MB neurons. We found that many neurons in V2 exhibited a robust orientation selectivity to MBs. The responses of V2 neurons to MBs also had a clear relationship with the animals' behavioral choice. As a comparison, these features were much weaker or absent in area V1. Finally, cross-correlogram and timing analysis also showed a potential functional link between DS and MB neurons.

## Results

Two macaque monkeys (monkey S and monkey W) were trained to do orientation-discrimination tasks. In each of their four hemispheres, a surgery was performed, during which intrinsic signal optical imaging was performed and a 32-channel array was implanted. The array was placed to cover as much V2 direction-preferred domains as possible (*Figure 1A–D*). The depths of the electrode tips were ~600 μm. For each array, recordings were performed daily and lasted for 1.5–2 months. In 128 channels of the four arrays, 96 channels were located in area V2 and 32 were in V1. Recordings were performed on multiple days. We used three alternative methods in cell selection (with different levels of strictness) and constructed three cell datasets: (1) 'all cell' dataset (n = 723). For daily recordings, a spike sorting was first performed for each channel, and signal-to-noise ratios were calculated for the resulting clusters (see Materials and methods). The best signal-to-noise ratio cluster for a channel was selected if its signal-to-noise ratio was larger than 3.5. With this method, we obtained 723 units from 85 V2 channels. (2) 'Unique-unit' dataset (n = 287). Based on the 'all cell' dataset, we excluded potential duplicated units (i.e., had similar waveforms and tunings) recorded from the same electrodes on different days so that the remaining units were 'unique' ones. This means that the neurons in this dataset were either from different electrodes or from the same electrode but had different waveform or tunings. (3) 'One-cell-per-channel' dataset (n = 85). In this dataset, only one neuron was chosen for each channel based on the 'all cell' dataset. Detailed cell identification procedures can be found in Materials and methods.

In the above cell identification, the majority of the neurons were single units (SUs, see Materials and methods). The percentages of multiunits (MUs) were 29.3%, 12.5%, and 27% in 'all cell', 'unique-unit', and 'one-cell-per-channel' datasets, respectively. We also performed data analysis with only SUs in each dataset and found that the inclusion of MUs does not change the main conclusions (not shown).

We compared the results obtained from these three datasets and found that the main results were the same. In order to exhibit more details of the data, the results presented below are based on the 'all cell' dataset. The results from the other two datasets are presented in *Figure 5—figure supplements 2* and *3* for comparison.

In daily recordings, we first performed basic RF tests when the monkey was doing a simple fixation, including cell RF mapping, orientation tests with sine-wave gratings, motion-direction tests with random dots (RDs), and MB orientation tests. The stimuli were 4° circular patches, covering the whole RF region of the array without being optimized for particular cells.

### V2 neurons selective to MB orientation

As shown in *Figure 1F* and *Video 1*, MB stimuli were composed with moving RDs. In the circular stimulus, RDs moved in opposite directions in the two sides of the virtual midline. The dots were the same on the two sides except for their moving directions. Thus, the boundary was mostly induced by motion. Similar to previous work (*Marcar et al., 2000*; *Chen et al., 2016*), two sets of MB stimuli were used in this study, in which the axes of the dot motion were either 45°-angle with the MB (MB1 stimuli) or 135°-angle with the MB (MB2 stimuli).

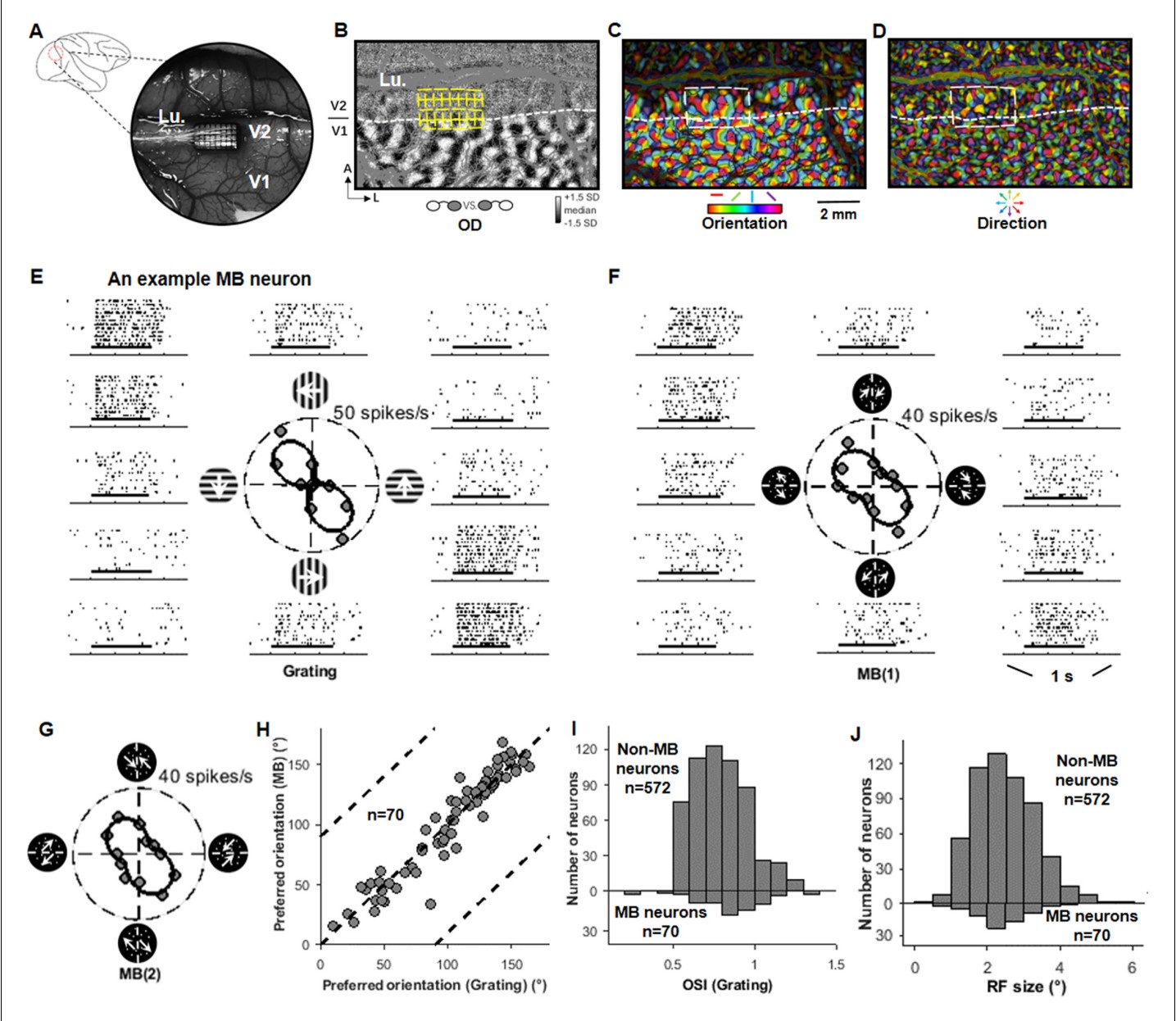

**Figure 1.** Map-guided array recording and V2 motion boundary (MB) neurons. (A) Illustration of an imaging chamber location and a 32-channel (4 × 8) array in an example case. Lu: lunate sulcus. (B) An ocular dominance map showing patterns in V1. The location of a 32-channel array is illustrated. (C) An orientation map for the same cortex region shown in (B). (D) A motion-direction map for the same cortex region shown in (B). (E) The responses of an example V2 neuron to sine-wave gratings presented in 12 drifting directions in 30° steps (six orientations). Short horizontal bars indicate stimulus presentation time (500 ms). (F) The responses of the same neuron to 12 MB1 stimuli (six orientations). In each MB1 stimulus, the moving axis of the random dots (RDs) was 45° clockwise from the midline. Each orientation was presented two times in which the RD-moving directions in the two sides were switched, plotted on opposite sides of the polar plot (also see *Figure 1—figure supplement 1E*). This neuron exhibited a strong MB orientation tuning, which was similar to its tuning to gratings (E). (G) Similar to (F), the responses of the same neuron to another set of MB stimuli (MB2), in which motion axis of the RDs had a different angle with the MB (135° clockwise from the midline). Also see *Figure 1—figure supplement 1E*. (H) For the 70 MB neurons, their preferred MB orientations were similar to their preferred grating orientations. (I) The distributions of the orientation selectivity index, measured with sine-wave gratings, for MB and non-MB neurons in V2. The MB neurons showed stronger orientation selectivity than the non-MB neurons (Wilcoxon test, p<0.01). (J) The distributions of the receptive field sizes for the MB neurons and the non-MB neurons in V2 were similar.

The online version of this article includes the following figure supplement(s) for figure 1:

**Figure supplement 1.** Additional information on array recordings.
**Figure supplement 2.** Array locations in relative to the V2 compartments.

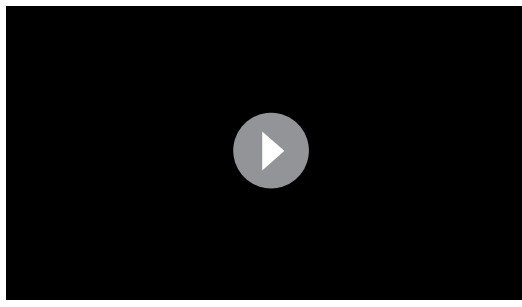

**Video 1.** Motion boundary (MB) stimulus for fixation task. The video shows the MB stimuli used to test neurons' MB orientation tuning during the fixation tasks. A MB stimulus is shown in two orthogonal orientations, each having two conditions in which the moving direction of the dots on the two sides was switched.

https://elifesciences.org/articles/61317#video1

Neurons were considered MB orientation selective (MB neurons) if their response functions were well fitted ($R^2$ >0.7), preferred orientations to the two MB stimulus sets were consistent (difference <30°), and both orientation selectivity indices (OSIs) for MB stimuli were larger than 0.5. According to these criteria, 10.9% (70/642) of V2 neurons were MB neurons (*Supplementary file 1*). This proportion is consistent with previous recordings in anesthetized monkey V2 (10.6%, 13/123, *Marcar et al., 2000*). Importantly, these MB neurons' preferred MB orientations were very similar to their preferred orientations for grating stimuli (*Figure 1H*). Thus, these MB neurons exhibited cue invariance for their orientation selectivity. *Figure 1I* shows that, when tested with grating stimuli, MB neurons exhibited stronger orientation selectivity than those non-MB neurons (mean OSI: 0.85 ± 0.20 vs. 0.80 ± 0.17, p=0.0013, Wilcoxon test). Their RF sizes, however, do not differ with the non-MB neurons' (2.38 ± 0.70° vs. 2.48 ± 0.87°, p=0.65, Wilcoxon test). Same for their response time delays (described in 'Response time courses').

These results were obtained in awake monkeys performing a fixation task and were very similar to those obtained in anesthetized monkey V2 (*Marcar et al., 2000*). Both studies showed that V2 has around 10% neurons that exhibit strong and robust orientation selectivity to both luminance and MBs, and their preferred orientation to these two stimuli is the same. These findings indicate that V2 has the capability to detect MBs and integrate this information with other boundary cues.

## Correlation between V2 activity and monkey behavior

To study whether V2 neurons contribute to MB perception, monkeys were trained to do an orientation-discrimination task. The task is illustrated in *Figure 2A*: after a sample MB stimulus was presented for 750 ms, two target dots were presented. The monkeys were required to indicate whether the MB line was tilted to the left or right of vertical by saccading to the left or right target. The difficulty of the task was adjusted by using different levels of motion coherence or dot brightness (in one chamber). After training, monkeys can perform well in MB orientation discrimination and achieved at least 95% correct rate for 100% coherence stimuli. *Figure 2B* shows the psychometric function (fitted with a Weibull function) for monkey S. The coherence thresholds, as determined at the 75% correct rate from the fitting function, were 36.7% for monkey S (*Figure 2B*) and 67.2% for monkey W (*Figure 2—figure supplement 1A*).

Based on the initial MB tests, we selected a MB cell for the following 'MB-orientation-discrimination test'. The stimuli were similar to the one used in previous MB tests, and its position was centered at the MB neuron's RF. Only two MB orientations were used either at the neuron's preferred orientation or orthogonal to that orientation. The stimuli were presented at different levels of difficulties (coherence or brightness). Recordings were made while the monkeys were performing an orientation-discrimination task. The responses of an example neuron are shown in *Figure 2C*. While the coherence of the moving dots increased, the differences of responses to preferred and null MB orientations also increased. *Figure 2D* shows the receiver operating characteristic (ROC) curves for four out of seven tested coherence levels for this neuron. Each curve indicates the difference of responses to preferred and null orientations. *Figure 2E* shows the example neuron's neurometric function, calculated based on the area sizes under the ROC curves. This curve is also well fitted by a Weibull function, and the threshold for this neuron is 40.1%. In our sampled neurons, this neuron was very sensitive to MB orientation. Its neurometric function and coherence threshold were both very similar to the monkey's behavior measurements.

Fifteen MB neurons were tested with coherence-level stimuli (*Figure 2F*). Their neurometric curves were generally shallower than the behavioral ones, and the neuronal thresholds were higher

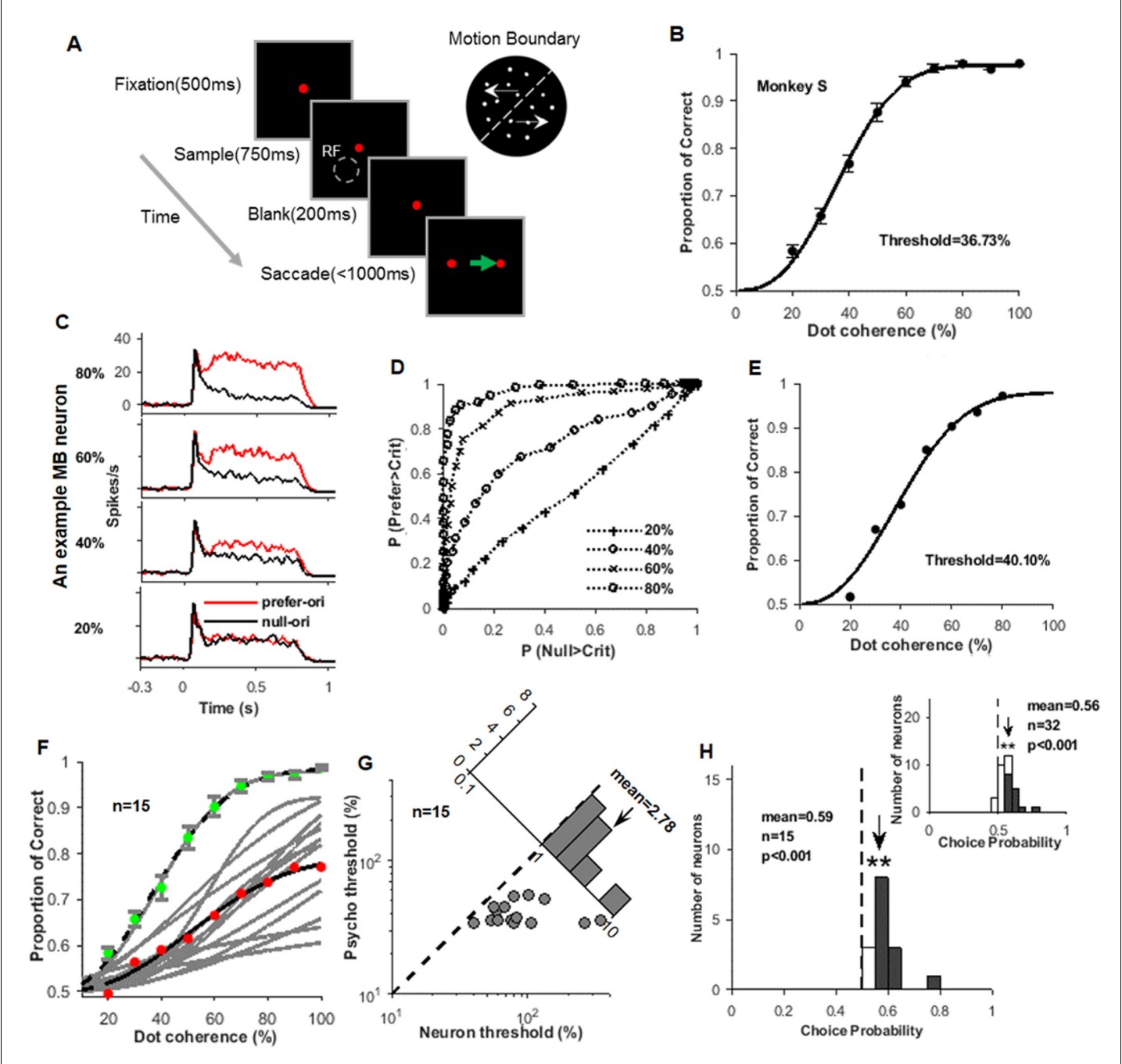

**Figure 2.** Animal performance and neural responses in the orientation-discrimination task. (**A**) An illustration of the motion boundary (MB)-orientation-discrimination task. A fixation point appeared for 500 ms, then a MB stimulus appeared for 750 ms. The fixation point maintained for additional 200 ms before it was replaced by two saccade targets. During this period, the monkey was required to maintain fixation at the fixation point. The monkey needed to make an eye saccade choice within 1000 ms, according to the tilt of the MB line. For example, for MB stimulus tilted to the right of vertical (as shown in the top right), the monkey should saccade to the right target. (**B**) The psychometric function of monkey S in the MB-orientation-discrimination task. (**C**) The response peri-stimulus time histograms (PSTHs) of an example MB neuron to MB stimuli presented different levels of motion coherence (four out of seven were shown), and the red and black curves are for the preferred and null MB orientations, respectively. (**D**) The receiver operating characteristic (ROC) curves obtained from the response PSTHs in (**C**), and horizontal and vertical coordinates are false alarm rate and hit rates, respectively. (**E**) The neurometric function obtained from the ROC curves in (**D**). (**F**) The neurometric functions (gray) of the 15 MB neurons recorded with the coherence stimuli. The red dots represent the average of the 15 neurometric functions, and the black line is its fitted curve. The green dots represent the average psychometric function measured during the recordings of the 15 neurons, and the black dashed line is its fitted curve. (**G**) Comparison of the neuronal thresholds and behavioral thresholds. The inset plot shows the distribution of ratios of the neural and behavioral

*Figure 2 continued on next page*

*Figure 2 continued*

thresholds. (H) The distribution of the choice probability (CP) values for the 15 MB neurons tested with the coherence stimuli, in which 12 neurons had CPs significantly larger than 0.5 (filled bars). The mean CP value (0.59) is also larger than 0.5 (t-test, p<0.001). The inset plot shows the distribution of CPs for all of the 32 neurons, including 15 tested with coherence stimuli (shown in the main panel) and 17 tested with brightness stimuli (also see *Figure 2—figure supplement 1B–E*).

The online version of this article includes the following figure supplement(s) for figure 2:

**Figure supplement 1.** Additional information for behavioral tests.

than the behavioral ones (*Figure 2G*). The average ratio of neuronal and behavioral thresholds was 2.78 (*Figure 2G*, inset plot). These results indicate that most neurons had a poorer sensitivity to MB orientation compared with the animals' behavioral performance. However, some most sensitive neurons had thresholds close to the behavioral ones. This indicates that at least some neurons had sufficient information to support the animals' performance in the MB orientation-discrimination tasks.

We further calculated the choice probability (CP) for these MB neurons (*Figure 2H*), which reflect the biases of neurons when their responses were grouped according to the monkeys' choices (*Britten et al., 1996*). In these neurons, 12 had a CP significantly larger than the chance level of 0.5 (bootstrap test, p<0.05). The average CP, 0.59, was also larger than 0.5 (t-test, p=$1.4 \times 10^{-5}$). This indicates that during the MB orientation-discrimination tasks MB neurons in V2 showed significant choice-related activity. In 17 neurons recorded in one array, we also tested neural-behavioral relevance with different levels of dot brightness. The overall CP was lower in neurons tested with dot brightness. This might be due to the relative easiness for the brightness task. Nevertheless, in both tasks, the average CPs were larger than 0.5 and portions of individual neurons had CPs values larger than 0.5 (*Figure 2—figure supplement 1B–E*). The CP distribution for the pooled neurons (n = 32, *Supplementary file 2*) is shown in *Figure 2H* inset, which also exhibit the same trend.

Thus, V2 not only contained neurons sensitive to MB orientation, the highest sensitive ones also exhibited orientation-discrimination performances that are close to the animal's behavioral performances. Their random response fluctuations correlated with the fluctuation animals made in the MB orientation-discrimination tasks. All these results indicate that V2 MB neurons likely contribute to the MB discrimination task.

## Comparison of MB responses in V1 and V2

In our sampled V2 population, 63% of the neurons were not tuned for MB orientation (either one or two MB responses could not be well fitted by the fitting function, i.e., $R^2$ <0.7), 26% either had a low OSI (<0.5) or their two preferred MB orientations were different (>30°), and the remaining 10.9% neurons were classified as MB neurons (*Figure 3A*, top). In our array recordings, 32 channels in three arrays were located in area V1 and from which 93 V1 neurons were recorded (*Figure 1—figure supplement 2*, *Figure 3—figure supplement 1*). Compared with V2, a higher percentage (82%) of V1 neurons were not tuned to MB orientation; 16% had a low OSI or their two preferred orientation differed; only 2% (two neurons) were classified as MB neurons (*Figure 3A*, bottom). V1 and V2 were different in their cell distributions in the three groups (chi-squared test, $\chi^2$=14.2, p=$8.25 \times 10^{-4}$). In addition, the mean orientation index (OSI) for MB stimuli was larger for V2 neurons than for V1 neurons (calculated from the neurons whose responses could be well fitted, i.e., $R^2 \geq 0.7$). Thus, V2 exhibited significantly higher MB detection capability than V1 at the population level.

In *Figure 3A*, we compared two OSIs for those well-fitted neurons, including both MB and non-MB neurons. V1 had an overall lower OSI value than that in V2 (V1: 0.36 ± 0.17; V2: 0.44 ± 0.19, p=0.033, Wilcoxon test). In both V1 and V2 neurons, their two OSIs were strongly correlated (V1: r = 0.70, p=0.0016; V2: r = 0.72, p=$3.9 \times 10^{-39}$). In addition, the OSI values of the MB and non-MB neurons were continuously distributed and did not form two clusters.

We further analyzed CPs for the whole V2 population and compared with those in V1. In this analysis, we also included both MB neurons and non-MB neurons. Most neurons recorded in an array had overlapped RFs (*Figure 1—figure supplement 1B*). In offline analysis, we identified non-MB neurons that had their RFs and preferred/null orientations matched the MB stimuli tested on that day and calculated their CP values in a similar way as for the MB neurons (*Figure 2H*). *Figure 3B* top panel shows the CP distribution for all V2 neurons suitable for CP analysis (n = 140), including 108

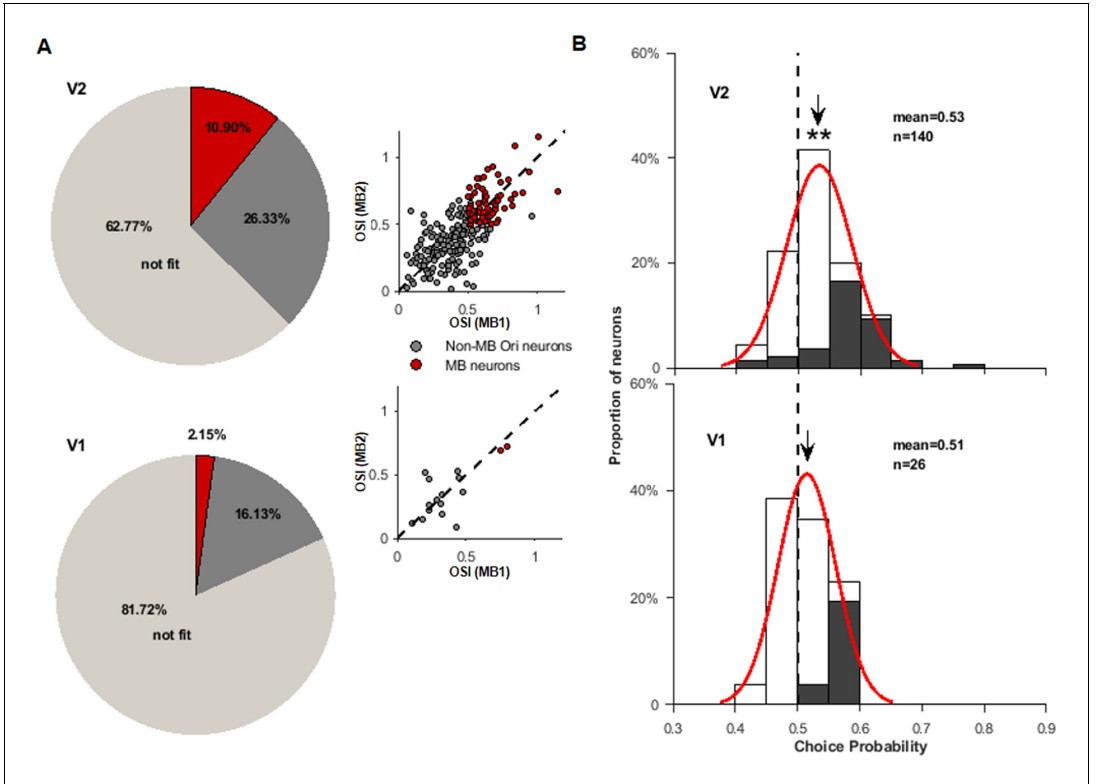

**Figure 3.** Comparison of V1 and V2 responses to motion boundary (MB) stimuli. (**A**) Two pie charts show the percentages of neurons in the recorded V2 (above) and V1 (below) neurons. Light gray represents neurons whose responses could not be fitted ($R^2<0.7$) by the von Mises function ('not fit'). Dark gray represents neurons that had their responses well fitted ($R^2 \geq 0.7$) but either had low orientation selectivity indices (<0.5) or the two preferred MB orientations did not match (differed by >30°). Red section represents MB neurons. On the right side of each pie chart, neurons' orientation indices for MB1 and MB2 stimuli are plotted. The colors are consistent with those in the pie charts, and dashed lines are the diagonal lines. (**B**) The distributions of choice probabilities (CPs) for all measured V2 (top) and V1 (bottom) neurons (including MB and non-MB neurons). For non-MB neurons, the preferred choice was based on their preferred orientations for gratings. Filled bars represent neurons having CP significantly different from 0.5.

The online version of this article includes the following figure supplement(s) for figure 3:

**Figure supplement 1.** Comparison of receptive fields (RFs) of V1 and V2 neurons.

**Figure supplement 2.** Choice probability (CP) distributions of motion boundary (MB) and non-MB neurons in V1 and V2.

non-MB neurons and 32 MB neurons. V2 CPs had a unimodal distribution and shifted toward the right side. Its mean CP (0.53) was higher than 0.5 (t-test, p=1.9 × 10$^{-12}$), but lower than the mean CP for MB neurons (0.56, *Figure 2H*). There were 49 V2 neurons that had a CP either larger or smaller than 0.5 (bootstrap test, p<0.05), among which 15 were MB neurons (all had CPs larger than 0.5). For the neurons recorded in V1, 26 were suitable for CP analysis, including 2 MB neurons (*Figure 3C*, bottom). The mean CP (0.51) was not different from 0.5 (t-test, p=0.11), and six neurons had a CP larger than 0.5 (bootstrap test, p<0.05). In addition, non-MB V2 neurons (n = 108) had a mean CP (0.53) larger than 0.5 (t-test, p<0.001), but not for non-MB V1 neurons (n = 24, CP = 0.51, p=0.21, t-test) (*Figure 3—figure supplement 2B, D*).

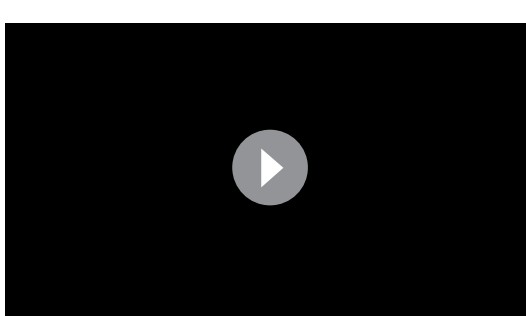

**Video 2.** Motion boundary (MB)-position stimuli. The video shows the MB-position stimuli used to test MB–DS correlation during the orientation-discrimination tasks. A MB was presented at different orientations and positions. DS: direction-selective.

https://elifesciences.org/articles/61317#video2

In summary, V2 not only had stronger orientation tuning to MB stimuli and more MB neurons than V1, their behavioral relevance was also stronger than the V1's.

## Correlations between MB and DS neurons

In our array recordings, we also tested neurons' direction selectivity with moving RDs. We identified 88 V2 DS neurons that had direction-selective index (DSI) larger than 0.5. In our hypothesis, V2 MB neurons receive their motion information from V2 DS neurons. In order to test this hypothesis, we examined the correlations between the MB and DS neurons for different types of visual stimuli.

The stimuli we used was a 4° square RD patch (*Figure 4—figure supplement 1A* and *Video 2*) that was divided into two parts by a virtual line (MB). The RD in one part moved in the DS neuron's preferred direction, and the RD in another part moved in its opposite direction (null direction). The square patch was rotated so that the MB matched the MB neuron's preferred/null orientations. We tested seven MB locations. The monkey was performing an orientation-discrimination task. We found that when the MB stimuli was presented cross-correlograms (CCGs) show an elevated correlation at 0 time lag (*Figure 4A*, data pooled for seven MB locations). Importantly, the correlation was much higher for the preferred MB orientation than the null one. Note that the moving directions of the RD were the same for these two conditions. We also analyzed DS–MB pairs (n = 5) in which DS neurons were not optimally stimulated. The CCGs for preferred MB and null MB do not differ (t-test, p=0.26). All CCGs we used were shuffle-corrected CCGs so that stimulus-related effects were removed.

We tested a luminance-line control stimulus (real line) (*Figure 4—figure supplement 1A* and *Video 3*), in which a white line replaced the MB, and in both sides the RD moved in two opposite directions (transparent motion). The correlation in CCG was reduced in these conditions and did not differ for the preferred and null orientations.

We further tested another version of control stimulus called temporal boundary (TB, *Chen et al., 2016*; *Figure 4—figure supplement 1A* and *Video 4*). TB stimuli were similar to the MB except that RDs in the two sides of the TB moved in the same direction. The dots still disappeared and appeared at the virtual boundaries. In examined neurons, none exhibited tuning to the TB orientation. The animals also made random choices to these stimuli in the orientation-discrimination tasks. In CCG, the MB–DS pairs exhibited larger modulation amplitudes to the TB stimuli (note the scale difference), but did not peak at any particular positions (*Figure 4C*). No differences were found for preferred and null orientations. The cortical distances of paired neurons in *Figure 4A–C* had a range of 0.93–1.1 mm.

We did the same analysis on pairs of non-MB and DS neurons. To make fair comparisons, we randomly selected equal numbers of non-MB–DS pairs as the MB–DS pairs (*Figure 4A–C*, right panel). Among all three stimulus conditions (MB, real line, TB), only under the MB condition that the non-MB–DS pairs exhibited small peaks at time zero, but did not differ between two orientations. To quantify the CCGs, we calculated the area sizes under the CCG curves from −40 ms to 40 ms. As shown in *Figure 4D*, only the MB–DS pairs under the MB condition showed significant differences between the preferred and null orientations (t-test, p=$1.9 \times 10^{-4}$).

In summary, results from response correlation analysis support the hypothesis that functional connections between V2 DS and MB neurons do exist. Such connections operate in specific stimulus conditions, but peak location (at time zero) is inconsistent with a simple DS to MB contribution model. These phenomena can also be due to bidirectional interactions between DS and MB neurons or their common inputs.

## Response time courses

To further investigate the mechanisms underlying V2 MB orientation selectivity, we measured the time courses of V2 MB neurons when they responded to the MB and real-line stimuli. Responses PSTHs (*Figure 5A, B*) were calculated from the data used in the correlation analysis above. In seven MB positions (*Figure 4—figure supplement 1A*), only the conditions that the MB was in the RF center (0°) were used in this analysis. With an analysis method similar to the one used in *Chen et al., 2017*, we calculated a visual response latency (i.e., the time a neuron started to respond to the visual stimulus) and an orientation-selectivity latency (i.e., the time responses started to differ for preferred and null orientations). The orientation-selectivity latency for the MB stimuli (85 ms, *Figure 5A*) was

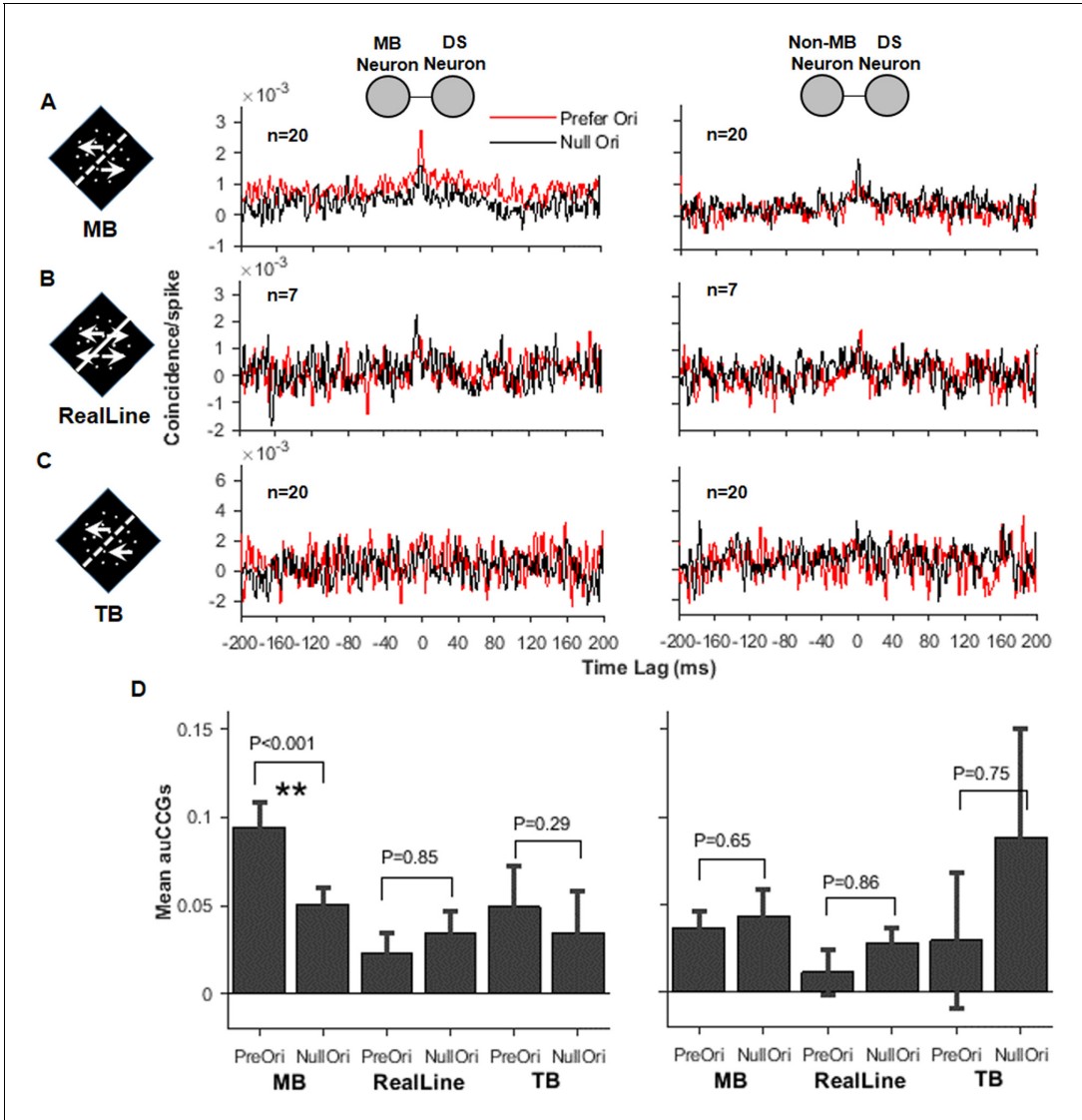

**Figure 4.** Correlated activity between the direction-selective (DS) and motion boundary (MB) neurons in V2. (**A**) Cross-correlograms (CCGs) for MB–DS pairs (left) and non-MB–DS pairs (right) in MB stimulus conditions. Red curves are for preferred orientations and black for null orientations. For non-MB neurons, MB stimuli were oriented along their preferred and null orientations for gratings. (**B**) Similar to (**A**), CCGs for real-line conditions. (**C**) Similar to (**A**) and (**B**), CCGs for temporal boundary conditions. (**D**) Accumulated CCGs for all stimulus conditions shown in (**A–C**). Only the MB–DS pairs in the MB conditions exhibited significantly higher correlation for the preferred orientation than the null one (t-test, p<0.001). Error bar: SEM.

The online version of this article includes the following figure supplement(s) for figure 4:

**Figure supplement 1.** Additional information for the correlation tests.

50 ms later than that for the real-line stimuli (35 ms, *Figure 5B*), while the visual response latencies for these two stimuli were similar (both were 37 ms, *Figure 5A, B*). This indicates that, compared with the luminance stimuli, additional time and circuits are required to calculate orientation from the MB stimuli. In addition, MB neurons had similar visual response latencies as those in non-MB neurons when they were responding to the MB stimuli (*Figure 5—figure supplement 1A, B*). This is different from the previous findings obtained in anesthetized monkeys (*Marcar et al., 2000*), where MB neurons responded slower to the MB stimuli than that of the non-MB neurons.

In order to see whether the response time of DS neurons falls into a reasonable range so that their contribution to MB neurons is valid, we analyzed the response time courses of V2 DS neurons (n = 9). Considering that V2 DS neurons have strong surround suppression (*Hu et al., 2018*), we tested three types of RD stimuli to stimulate their RF centers and surrounds: (1) a center-only RD

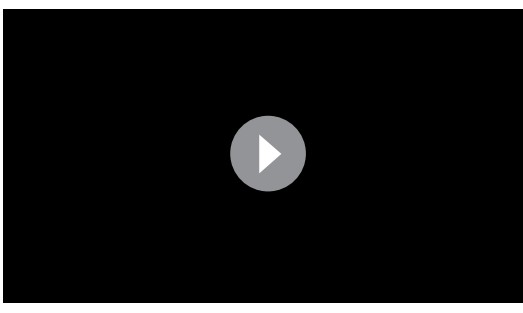

**Video 3.** Real-line position stimuli. The video shows the real-line stimuli as control for the motion boundary-position stimuli.

https://elifesciences.org/articles/61317#video3

patch in which RDs moved in the neurons' preferred directions; (2) a large RD patch covered both RF center and surround in which RDs moved in the neurons' preferred directions; and (3) a center patch moved in the preferred directions and a surround patch (annulus) moved in the opposite (null) directions. Latencies were calculated with a population latency method (see Materials and methods). The visual response latency for DS neurons was 41 ms, similar for all three stimuli. The surround suppression time started at 51 ms, when responses to stimuli 1 and 2 started to differ. The surround modulation time started at 109 ms, when responses to stimuli 2 and 3 started to differ. These time points, together with those from MB neurons, are summarized in *Figure 5D*. Since we had only tested a limited number of DS neurons, and there are also large variations among DS neurons, these values may not be very precise. Nevertheless, what we can see is that the emergence of MB orientation selectivity (85 ms) roughly falls in the time range (51–109 ms) of surround effects operated in the DS neurons. This indicates that DS neurons have the potential to contribute to the MB orientation selectivity in MB neurons. In addition to the population latency method, we also calculated latencies for individual neurons, thus the variance can be evaluated (*Figure 5—figure supplement 1C, D*). The temporal relationship between MB and DS responses calculated from these two methods was consistent.

## Discussion

We studied MB neurons in awake monkeys performing an orientation-discrimination task. There were 10.9% V2 neurons that exhibited robust, cue-invariant orientation selectivity to MB and luminance stimuli. V2 neurons also exhibited a correlation with animals' MB orientation-discrimination behavior. Compared with V2, V1 had fewer MB neurons and a much weaker correlation with animals' MB orientation-discrimination behavior. Evidence from temporal correlation between MB and DS neurons, as well as their response latencies, supports the model in which MB neurons receive input from DS neurons during MB orientation detection.

These results are largely consistent with previous findings on anesthetized monkeys (*Marcar et al., 2000*), which indicate that V2 is important for MB detection. With array recordings on awake behaving monkey, we provided novel evidence showing that (1) V2 MB neurons may contribute to MB perception and (2) V2 MB detection may use motion information within V2.

### MB orientation selectivity in V2

A moving object contains much information that can aid object recognition, including figure-ground segregation (*Lamme, 1995*) and contour recognition (*Regan, 1986*). Compared with figure-ground segregation, contour recognition may require more precise location of the MB and more complicated boundary integration process. A major finding of this work is the correlation between V2 responses and behavioral MB perception. This finding supports the view that V2 is a key area in MB perception. It is also in line with findings in previous loss-of-function studies. For example, monkeys are unable to detect MB after a V2 lesion (*Merigan et al., 1993*), and human patients with V2 lesions are

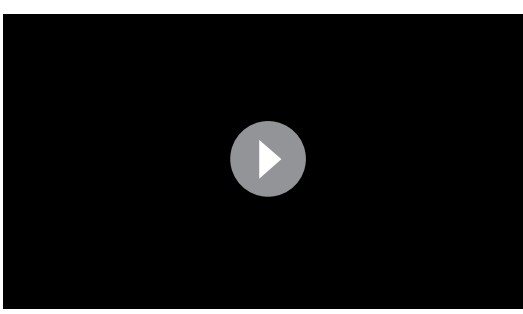

**Video 4.** Temporal boundary (TB) stimuli. The video shows the TB stimuli used as a control for the motion boundary-position stimuli.

https://elifesciences.org/articles/61317#video4

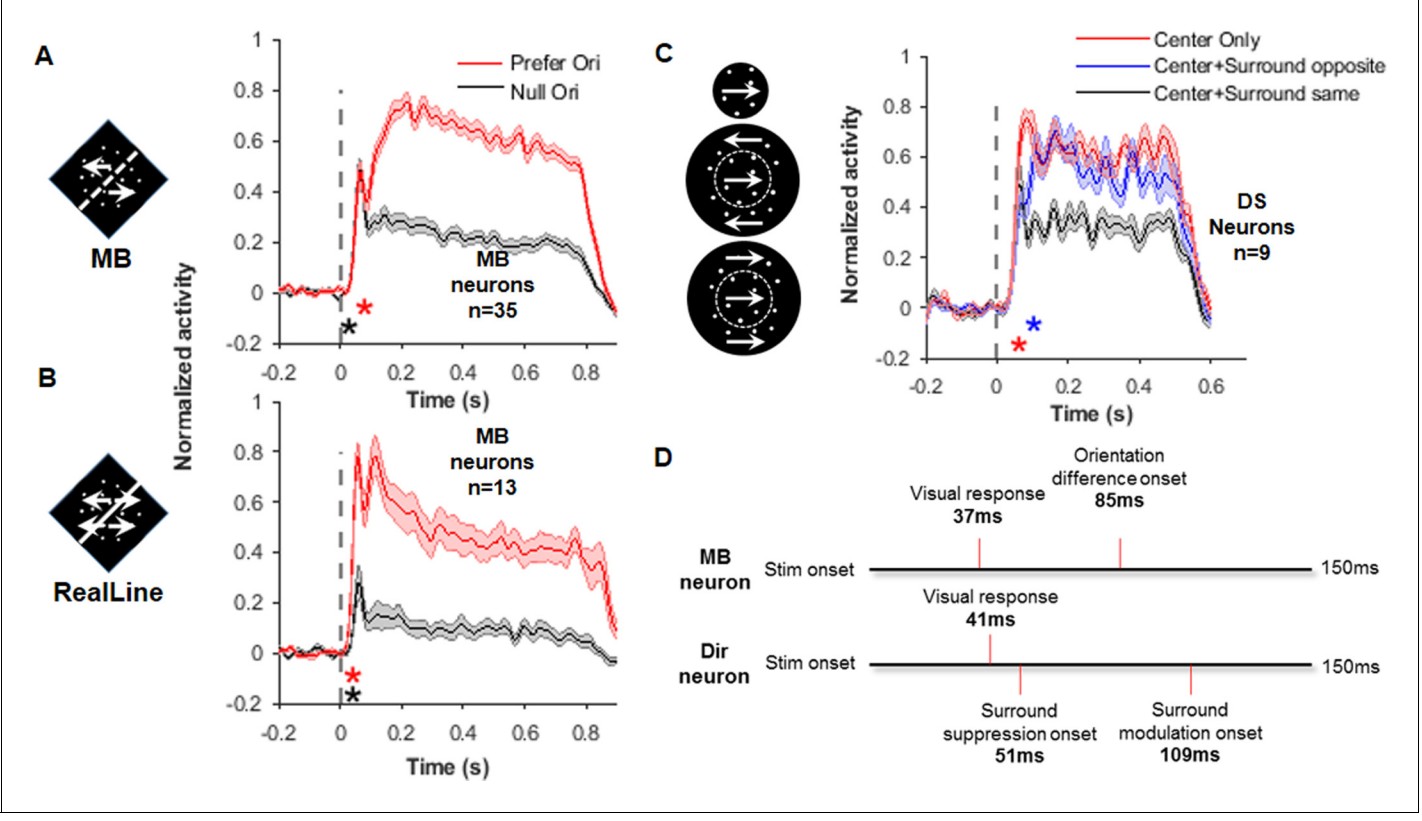

**Figure 5.** Response time courses of the motion boundary (MB) and direction-selective (DS) neurons. (**A**) Average response PSTHs for the MB neurons to the MB stimuli presented at their preferred (red) and null orientations. The black asterisk indicates the time for visual response delay (37 ms), which was similar for these two stimulus conditions. This value was also similar to that in non-MB neurons for the same stimuli (39 ms, see *Figure 5—figure supplement 1A*). The red asterisk indicates the time point (85 ms) where responses to the preferred and null orientations started to show differences (i. e., MB orientation time). Error shading represents SEM, same for below. (**B**) Average response PSTHs for the MB neurons to the real-line stimuli. The two asterisks indicate the visual response delay (37 ms, black asterisk) and the orientation-selective response time (35 ms, red asterisk), respectively. Non-MB neurons' responses to the real line were similar (*Figure 5—figure supplement 1B*). (**C**) Average response PSTHs for DS neurons in responding to three different stimuli: center only (red), center and surround with the same direction (black), and center and surround with opposite directions (blue). The center random dots were always moving at the DS cell's preferred direction in these three conditions. The red asterisk indicates time when the 'center surround same' responses start to different from the 'center-only' responses. The blue asterisk indicates the time where the 'center surround opposite' responses started to different from the 'center surround same' responses. (**D**) A summary of time points shown in (**A–C**). The time MB neurons start to show MB orientation selectivity (85 ms) falls in the time range where surround effects emerge in the DS neurons.

The online version of this article includes the following figure supplement(s) for figure 5:

**Figure supplement 1.** Additional information for response delay.

**Figure supplement 2.** Results from 'one-cell-per-channel' dataset.

**Figure supplement 3.** Main results analyzed based-on 'unique-units' dataset.

**Figure supplement 4.** Two alternative direction-selective to motion boundary (DS-to-MB) models in which surround suppression of DS neurons plays different roles.

unable to detect motion discontinuity (*Vaina et al., 2000*).

In the distributions of OSI for MB stimuli (*Figure 3A*), we found that MB neurons and non-MB neurons do not form two separate clusters, and neurons with different levels of MB orientation selectivity were all observed. Also, the mean CP for non-MB V2 neurons is higher than chance level 0.5 (*Figure 3—figure supplement 2B*). Thus, the contribution V2 makes to MB perception is likely population-coding based and not limited to the strongest MB neurons. Through pooling, a stronger and more robust MB signal can be achieved in V2 at the population level or in the downstream boundary detection units. For example, a weighted summation model (*Jazayeri and Movshon, 2006*) can be used in MB detection task. In this model, a likelihood function (i.e., a probability density function) is calculated based on the weighted-summing of all activated neurons (optimally and suboptimally).

Based on the likelihood function, a series of behavioral tasks (e.g., detection, discrimination) can be achieved. Our previous work shows that V2 contained a strong orientation map for MBs (*Chen et al., 2017*); that map may represent a summation of various levels of such contributions.

Compared with V1, V2 improves its MB detection in at least three aspects: the percentages of MB neurons, functional mapping signal obtained with optical imaging (*Chen et al., 2016*), and correlations with MB perception. Nevertheless, such improvement is at population level. Although fewer, V1 does have MB neurons and neurons significantly correlated with animals' MB perception. Such a V1–V2 difference was also observed in other second-order boundary responses (e.g., to illusory contours). A better understanding of neural mechanisms underlying MB detection can be achieved through correlation and time-course analysis on V1 neurons, which were not performed in this study due to limited cell numbers.

Our samples of V1/V2 neurons were limited to the regions close to the V1/V2 border. This might affect our results in several aspects. For example, although the ocular dominance maps show clear V1/V2 borders, the actual transition from V1 neurons to V2 neurons was unlikely that sharp, especially for neurons in superficial layers. There should be a narrow 'transition zone' in which V1 and V2 neurons were actually mixed or inseparable. Plus, there were always precision limits in our map-array alignments. Thus, it is possible that some of our V1 neurons were actually V2 neurons, or the other way around. The RF size of V1 neurons we measured was relatively large (2.5° ± 1.18°) and might be due to this factor. We analyzed a subset of V1 cells (n = 39) that were recorded from the electrodes further away from the V1/V2 border and found that their mean RF size was smaller (2.05°). The possible partial 'mixture' of V1 and V2 neurons in our samples would decrease the differences between the V1 and V2 population, which means the actual differences might be larger than what we observed. In addition, the relatively large stimulus size (0.8°) used in RF mapping, as well as the eye movements in awake recording, might also contribute to the relatively large V1 RF we measured. However, since the stimulus sizes were not adjusted based on the overestimated RFs, the RF centers were correctly measured (*Figure 3—figure supplement 1*), and the RFs were covered by a relatively large stimuli (4°), we think the larger V1 RF sizes we measured should not affect the main results we observed (*Figure 3*).

To isolate and study motion-induced boundary perception, we used a very specialized and unnatural stimulus, which contained only motion-direction contrast. Other motion-related aspects, for example, speed contrast, were not examined. Beside this, real object boundaries usually contain multiple cues (luminance, texture, color, disparity, motion, etc.). Previous studies (*Peterhans and Heydt, 1993*; *Leventhal et al., 1998*; *Chen et al., 2017*; *El-Shamayleh and Movshon, 2011*; *von der Heydt et al., 2000*) have found V2 neurons that respond selectively to these boundaries and their orientations. Portions of V2 neurons (*Peterhans and Heydt, 1993*; *El-Shamayleh and Movshon, 2011*; *Marcar et al., 2000*) also exhibit cue-invariant responses to these boundaries. Thus, our results join those previous findings to show that V2 is a key area for boundary detection and integration.

Considering CO histology is unavailable yet for these monkeys, we estimated stripe types based on optical imaging maps (ocular dominance, orientation, color, motion direction). To separate thick and pale stripes, we used a width ratio previously described (thick:pale = 2:1.5, *Shipp and Zeki, 1989*). *Figure 1—figure supplement 2I–L* shows the locations of the four arrays overlaid on the corresponding color versus luminance maps. *Figure 1—figure supplement 2M* shows neuron numbers according to their stripe and tuning types. Although a statistical analysis was not available due to the overlapped tuning types, one can still draw some conclusions from the table. For example, DS neurons were more likely found in thick stripes, while MB neurons were more likely found in thick and pale stripes. These results were all consistent with previous findings (*Hubel and Livingstone, 1987*; *Levitt et al., 1994*; *Shipp and Zeki, 2002*; *Lu et al., 2010*; *Chen et al., 2016*).

## MB coding mechanisms

Extracting boundary information from motion is important for object recognition (*Regan, 1986*; *Regan, 1989*), especially when other cues are unavailable or confusing (e.g., camouflage). In primates, MB orientation extraction does not occur in the dorsal pathway (*Marcar et al., 1995*), where abundant motion information is available. Instead, this occurs in the ventral pathway and in which V2 plays a critical role. Two fundamental questions regarding the neural mechanisms underlying MB orientation detection are (1) the origin of motion signal for the MB calculation in the ventral areas. It

can come from dorsal areas through dorsal–ventral connections or originate from within the ventral stream itself. (2) Whether the MB calculation occurs in V2? And if it does, how is it calculated? If it is not in V2, then where V2 obtains this ability?

Our hypothesis is that V2 is an important area calculating MB orientation, and it also uses its own motion signals for this process. The main supportive evidence is from previous work (*Chen et al., 2016*; *Hu et al., 2018*) and the present study. These studies found that (1) V2 has abundant resources for the MB calculation, including a large number of DS neurons, which cluster and form functional domains. A biological system normally uses a simple solution instead of a complicated one if both achieve the same result. To calculate MB locally is apparently a simpler solution than to calculate it elsewhere and send it to V2. (2) The motion signal in V2 is especially suitable for MB calculation. V2 DS neurons have small RFs and a high motion contrast sensitivity (*Hu et al., 2018*). From their V1 inputs, V2 DS neurons retain and strengthen their center-surround modulation (*Hu et al., 2018*). Its functional 'purpose' is probably for MB detection. (3) The MB–DS functional correlation and time-course orders also support this hypothesis. Further evidence on correlation analysis, although did not reach statistical significance, also supports this hypothesis (*Figure 4—figure supplement 1C*). (4) MB is a useful signal that presents everywhere in the dynamic visual world. V2 integrates boundary information from various types of cues (luminance, color, disparity, texture, etc.). An early processing and integration of these cues is beneficial and simplifies the downstream processes.

We further suggest that the center-surround mechanism of DS neurons plays a significant role in MB detection. Our previous work has shown that DS neurons were activated optimally when a MB line touched the edge of their RF center (Figure 5 in *Hu et al., 2018*). In addition, the precise location of the MB line was also detected at the same time (*Figure 5—figure supplement 4*, model 1). Theoretically, MB detection can also be achieved without relying on the center-surround mechanism of DS neurons. For example, by comparing the activation of DS neurons signaling the motion of the two moving objects (not the motion contrast) (*Figure 5—figure supplement 4*, model 2). The onset time of direction selectivity for DS neuron (49 ms, *Figure 5—figure supplement 1E*) was faster than the MB-selectivity time of MB neurons (85 ms), and thus was not against such models. However, we have shown that DS neurons are suboptimally activated in such conditions (Figure 5 in *Hu et al., 2018*) and a precise MB location is unavailable. Thus, a center-surround mechanism is more suitable for this task. In the primate visual system, such a center-surround mechanism appears to be used as a general strategy in detecting changes of first-order cues at edges (e.g., luminance, color, disparity, etc.).

This hypothesis does not mean to exclude the contribution of feedback (e.g., from V3 or V4). Feedback plays a role in many V2 processes, and very likely in MB process as well. A supportive evidence for feedback is that the both MB and non-MB cells had a longer stimulus onset latency for MB stimulus than for luminance stimulus (*Marcar et al., 2000*). However, the result was obtained from anesthetized monkeys. Our recordings from awake monkeys did not show such difference (*Figure 5A*, *Figure 5—figure supplement 1A, B*). It is also possible that V2 and higher-level areas both do MB extraction but operate under different stimulus conditions. Higher-level areas may calculate MB for larger-scale stimuli or weaker motion contrast (thus requires larger spatial summation).

In an earlier study, it has been found that about 34% V2 neurons have orientation selectivity for coherently moving line of dots (*Peterhans and Heydt, 1993*; *Peterhans et al., 2005*). The neural mechanisms underlying these two types of orientation-from-motion selectivity, however, might be different. The MB detection we described is mainly based on the motion contrast between the figure and ground, while theirs relies on the coherent motion of the dots. The percentages of neurons were also different. Nevertheless, the neurons they described also performed orientation detection based on motion information. Both studies show the importance of motion information in boundary detection in V2.

In addition to basic orientation selectivity, V2 neurons respond to more complex contours (*Kobatake and Tanaka, 1994*; *Hegdé and Van Essen, 2007*; *Ito and Komatsu, 2004*; *Anzai et al., 2007*). A significant feature distinguishing V2 from V1 is its cue-invariant contour detection, for example, in illusory contours (*Von der Heydt and Peterhans, 1989*), MB (*Marcar et al., 2000*; *Yin et al., 2015*; *Chen et al., 2016*), disparity-defined contours (*von der Heydt et al., 2000*; *Bredfeldt and Cumming, 2006*), and possibly texture-defined contours (*El-Shamayleh and Movshon, 2011*). In addition, V2 shows border-ownership features (*Zhou et al., 2000*). These findings

suggest that figure-ground segregation is an emphasized visual process in area V2. The neural mechanisms underlying these processes remain unclear. Our work made an effort in examining the neural mechanisms underlying the MB detection in area V2.

For different types of contour cues, a common strategy can be used to extract contrast and then orientation. That is, a three-stage process (cue cell – contrast cell – boundary orientation cell). In this process, different cue cells involve in the first two stages, for example, DS neurons for motion contrast extraction (*Hu et al., 2018*), or relative disparity neurons for disparity contrast (*Thomas et al., 2002*). Different information may converge to common third-stage neurons for cue-invariant orientation detection, which not only enhances orientation sensitivity but also is more efficient.

## Functional contribution of V2 motion processing

Although there are abundant DS neurons in V2 (17.5%, *Hu et al., 2018*), the perceptual contributions of these DS neurons are still unknown. These DS neurons' RF properties are quite different from those in MT (*Hu et al., 2018*). Neither do they project to MT (*Ponce et al., 2008*; *Ponce et al., 2011*). Their sensitivity to motion contrast makes them suitable for detection of motion parallax, and their small RFs provide precise locations of the motion contrast. To our knowledge, the MB–DS correlation we observed provides the first direct evidence showing what perceptual function V2 DS neurons may contribute to. Besides MB detection, the contributions of V2 DS neurons to other perceptual tasks, like biological motion, and 3D-structure-from-motion, remain to be explored.

# Materials and methods

## Key resources table

| Reagent type (species) or resource | Designation | Source or reference | Identifiers | Additional information |
|---|---|---|---|---|
| Strain, strain background (macaque, male) | *Macaca mulatta* | Suzhou Xishan Zhongke animal Company, Ltd | NCBITaxon:9544 | http://xsdw.bioon.com.cn/ |
| Software, algorithm | MATLAB-R2018b | MathWorks | SCR_001622 | R2018b |
| Software, algorithm | Codes and 'all cell' dataset | This paper | N/A | http://dx.doi.org/10.17632/fjy37kc8pd.3 |
| Software, algorithm | Codes and unique-unit dataset | This paper | N/A | http://dx.doi.org/10.17632/fjy37kc8pd.3 |
| Software, algorithm | Codes and 'one-cell-per-channel' dataset | This paper | N/A | http://dx.doi.org/10.17632/fjy37kc8pd.3 |
| Other | 32-channel-array | Blackrock Microsystems, LLC | N/A | https://www.blackrockmicro.com/ |
| Other | 64-channel Multichannel Neural Recording (AlphaLab SNR) | Alpha Omega | N/A | https://www.alphaomega-eng.com/ |
| Other | Imager 3001 (Optical Imaging) | Optical Imaging Ltd | N/A | https://optimaging.com/ |
| Other | Eyelink Desktop (eyelink 1000) | SR Research | N/A | https://www.srresearch.com/ |

## Experimental overview

Monkeys were trained to perform orientation-discrimination tasks after headpost implant. After the training, a surgery was performed, during which optical imaging was made and an array was implanted. After 1 week of recovery time, awake electrophysiological recordings were made daily and usually lasted 1.5–2 months.

## Visual stimuli for optical imaging

In order to identify area V2 and place the electrode arrays over the direction-preference domains, we performed optical imaging before the array implants. The visual stimuli used were the same as

described previously (*Lu et al., 2010*; *Chen et al., 2016*). Briefly, we used square-wave gratings (SF: 1.5 c/deg, TF: 8 Hz) for mapping ocular dominance patterns in V1 (e.g., *Figure 1B*) and orientation maps in both V1 and V2 (e.g., *Figure 1C*). We used red–green isoluminant sine-wave gratings and black–white sine-wave gratings (SF: 0.15 c/deg, TF: 1 Hz, orientation: 45° and 135°) for mapping color/luminance patches in both V1 and V2 (e.g., *Figure 1—figure supplement 2I–L*). We used moving RDs (density: 3%, dot size: 0.1°, speed: 6°/s, monocularly presented) for mapping direction-preference maps in V2 (e.g., *Figure 1D*).

All visual stimuli were generated in ViSaGe (Cambridge Research System Ltd) with MATLAB scripts and were presented on a 21-inch CRT display (Dell P1130). The CRT display had a screen sized 80 × 60 cm, refreshed at 100 Hz, with maximum luminance (white) of 97.8 cd/m$^2$ and minimum (black) luminance of 0.03 cd/m$^2$. The distance between the CRT screen and the animal eyes was 57 cm.

## Visual stimuli for electrophysiology

Similar to imaging stimuli, stimuli for electrophysiological recordings were also generated with ViSaGe using MATLAB scripts and presented on a 21-inch CRT display (SONY CPD-G520). The CRT screen was 80 × 60 cm, refreshed at 100 Hz, with maximum luminance (white) of 56.7 cd/m$^2$ and minimum (black) luminance of 0.04 cd/m$^2$. The distance between the CRT screen and the animal eyes was 71 cm. The size of the stimulus was either a 4°-diameter circular, or a 4 × 4° square, except otherwise described. Stimulus presentation times were set for 500 ms (for the fixation tasks) or 750 ms (for the orientation-discrimination tasks).

## Moving gratings

Luminance sine-wave gratings were used to measure the orientation selectivity of neurons in the fixation tasks. The stimulus was a 4° circular patch, with SF of 2 c/deg and TF of 8 Hz (*Hu et al., 2018*). The grating patch had a mean luminance of 28.4 cd/m$^2$, 100% contrast, and drifted at 12 different directions (in 30° steps). Background luminance was 28.4 cd/m$^2$. Totally 12 conditions were tested.

## RF mapping

Two types of stimuli were used for RF mapping in fixation tasks. The stimulus was placed based on initial manual mapping. Depending on the array locations in V2, the centers of the stimuli were 0.3–0.8° from the vertical meridian horizontally and 2.3–3.1° below the fixation spot.

The first RF mapping stimulus was a 0.8° square-wave patch presented in a 4 × 4° square regions. Totally 25 possible locations in a 5 × 5 grid were tested (*Hu et al., 2018*). The gratings had an SF of 2 c/deg and TF of 8 Hz, and a duty cycle of 0.5; background luminance was 28.4 cd/m$^2$.

The second RF mapping stimulus was a white bar (56.7 cd/m$^2$), presented either vertically (for X-axis mapping) or horizontally (for Y-axis mapping) at different positions. The bar was 4° long, 0.2° wide. After determining the approximate RF locations, 21 positions were tested along the X-axis (vertical bar) and the Y-axis (horizontal bar) randomly, centered at the RF center location (10 positions on each side). Each position was separated by 0.2°. Background luminance was 28.4 cd/m$^2$. There were totally 42 conditions in this stimulus set.

## Random dots

Moving RDs were used to test the direction selectivity of neurons in the fixation tasks. RDs moved coherently in a 4° circular patch. Each dot was a pixel (0.04°), moved at 2°/s. Dot density was 5% (white dots cover 5% of the patch surface). The luminance of the dots and the background screen were white (56.7 cd/m$^2$) and black (0.04 cd/m$^2$), respectively. There were 12 moving directions (conditions) with a 30° interval.

## MB stimuli

The MB stimuli were used to test neurons' MB orientation selectivity in the fixation tasks. The stimuli are also illustrated in *Figure 1—figure supplement 1E* and *Video 1*. Similar to those used in previous studies (e.g., *Marcar et al., 2000*), each MB stimulus was a 4° circular patch of RDs. The patch was divided into two halves by a virtual border in the middle. The dots in the two halves moved in opposite directions, along an axis either 45° angle (MB1) or 135° angle (MB2) to the midline. The

other features of the dots on the two sides were the same, thus the MB in the middle only was visible when the dots were moving. Dots were 1 pixel in size (0.04°), moved at 2°/s, and had a density of 5%. The luminance of the dots and the background screen were the same with parameters of RDs. Six MB orientations, with an interval of 30°, were tested. For each MB orientation, there were four different conditions, including two dot-moving axes (MB1 and MB2) and two conditions in which dots on the two sides were switched. Thus, there were 24 conditions in this stimulus set.

### TB stimuli

The TB stimulus was used as a control for the MB stimulus in the fixation tasks. It was similar to the MB except that the dots on the two sides moved in the same direction. The dots still disappear or appear at the virtual border and created a weak boundary perception. Thus, the difference between TB and MB was that TB lacked motion contrast. Same as the MB stimuli, the TB stimuli had 24 conditions.

### MB orientation-discrimination stimuli

This stimulus set was used in orientation-discrimination tasks (e.g., *Figure 2*). The stimuli were similar to the 'MB stimuli' used in fixation tasks (described above), except they used seven levels of dot motion coherence or dot brightness. Typical values for the seven coherence levels were 65%, 70%, 75%, 80%, 85%, 90%, and 100% (for monkey W); and 20%, 30%, 40%, 50%, 60%, 70%, and 80% (for monkey S). Typical values for the seven brightness levels were 1.17, 2.87, 5.70, 11.36, 17.03, 22.69, and 28.35 cd/m$^2$ (only tested on monkey S). There were 56 conditions in this stimulus set (seven difficulty levels, two orientations, two dot-moving axes [45° and 135°], two dot-moving directions [toward/away from the MB]). Each stimulus condition was repeated 20–30 times. The average correct rate for this stimulus set was 80%.

### MB-position stimuli

This stimulus set was used to evaluate the correlation between neuron pairs and their time courses (*Figures 4* and *5*) and tested when monkeys were performing an orientation-discrimination task. An illustration of the stimulus is shown in *Figure 4—figure supplement 1* and *Video 2*. The stimulus was a 4 × 4° square of RDs, centered at the RF center of a selected MB neuron. The square was rotated so that its MB was oriented either at the MB neuron's preferred orientation or its null orientation. Seven MB locations were tested, which were −1.25°, −1°, −0.5°, 0°, 0.5°, 1°, and 1.25° away from the neuron's RF center, respectively. The moving directions of the RDs on each side of the MB were set at the preferred and null directions of a chosen direction-selective neuron. Other parameters of the RDs were the same as those in the 'MB stimuli' in fixation tasks. This stimulus set had 28 conditions. Each stimulus condition was repeated 30–50 times.

### Real-line stimuli

Real-line stimuli were used as control stimuli for the 'MB-position stimuli'. It has a white line at the original MB location. The white line was 4° long, 0.1° thick, and had the same luminance level as the RDs (56.7 cd/m$^2$). The RDs in both sides of the line were doing transparent motion, in preferred and null directions. Other parameters were the same as the 'MB-position stimuli' described above. This stimulus set had 14 conditions. Each stimulus condition was repeated 30–50 times. One hemisphere from each monkey was tested with this type of stimuli.

### TB-position stimuli

This stimulus set was also a control stimulus for the 'MB-position' stimulus and had a similar form. The RDs in the square patch moved in the same direction (as in the 'TB stimuli'), either at the neuron's preferred direction or null direction. Two TB orientations, either at the MB neuron's preferred orientation or null orientation, were tested. This stimulus set had four conditions (only center conditions). Each stimulus condition was repeated 30–50 times.

### Surround modulation stimuli

This stimulus set was used to evaluate the surround modulation effects in DS neurons and was used in fixation tasks. Similar to those used in previous studies (*Hu et al., 2018*), the RD patches were

divided into center and surround patches. The center patch was a circular one with size matching the cell's RF and had RDs moving in the neuron's preferred direction. The surround patch was an annulus surrounding the center patch with an outside diameter of 6°. RDs in the surround patch moved in one of eight directions covering the 360° (interval 45°) starting at the neuron's preferred direction. This stimulus set had nine conditions, including eight conditions for different surround directions and a center-alone condition. Similar to previous stimuli, the dots were 1 pixel in size (0.04°), moved at 2°/s, and had a density of 5%.

## Behavioral tasks
### Fixation task
Monkeys were trained to maintain fixation during stimulus presentation. Eye position was monitored with an infrared eye tracker (EyeLink 1000, SR Research). Each trial started with the appearance of a red fixation spot (0.2°) in the middle of the screen. There were two stimuli presented sequentially in each trial. After the monkeys maintained 500 ms fixation, the first visual stimulus appeared and lasted for 500 ms. The second visual stimulus appeared after a 250 ms interval and also lasted for 500 ms. The monkeys received a water reward (~0.1 ml) after having successfully maintained fixation during this whole period. The fixation window was typically 1.2 × 1.2°. The trial was aborted if the animal lost fixation at any time during the trial, and the affected stimulus data were discarded (the unaffected stimulus data were still used). The intertrial interval was 1 s (for correct trials) or 1.5 s (for failed trials). Each stimulus condition was usually repeated at least 10 times, except for the RF mapping stimuli, which were repeated minimum of five times.

## MB orientation-discrimination task
This task was a two-alternative forced-choice discrimination based on the MB orientation (illustrated in *Figure 2A*). The following visual stimulus sets were used in this task: 'MB orientation-discrimination stimuli', 'MB-position stimuli', 'real-line stimuli', and 'TB-position stimuli'. After a 500 ms fixation time, visual stimulus was presented for 750 ms (700–850 ms, see below). The fixation spot stayed for an additional 200 ms and was replaced with two saccade targets, which were 0.2° red spots 3.5° away from the center on both sides of the screen. The animals were required to make an eye saccade choice within 1000 ms: to saccade to the right target if the MB line was tilted to the right of vertical or to left if the MB line was tilted to the left of vertical. Monkeys received a water reward for correct choice. Interstimulus interval was 1 s for correct trials and 1.5 s for failed ones. For technical limitations, the actual stimulus presentation time had a variation from trial to trial (700–850 ms). The actual stimulus presentation times were recorded with a photo diode attached to the CRT screen and aligned in the offline data analysis.

## Surgery
All surgical procedures were performed under sterile conditions and general anesthesia. Similar to the surgery procedures described previously (*Li et al., 2013*; *Hu et al., 2018*), a 20–21 cm circular craniotomy was made above areas V1 and V2 (center location: 17–18 mm anterior to the posterior bone ridge and 18–20 mm lateral to the midline). The dura was retracted and the cortex was stabilized with 3–4% agar and covered with a 18–20 mm diameter cover glass. Intrinsic signal optical imaging was performed in order to determine V1/V2 border and V2 functional maps (described below).

The cover glass and agar were carefully removed when the imaging was completed. Array implant locations were selected based on the mapping results, usually centered at a V2 DS domain (*Figure 1A–D*). All arrays used were 32-channel (4 × 8) Utah arrays. Electrode lengths were 1 mm, with a spacing of 0.4 mm, and impedance of 500 kΩ. The electrode tips were inserted to ~0.6 mm below the surface. The craniotomy was covered with dental cements, and the animals were recovered.

## Optical imaging
In array implant experiments, intrinsic signal optical imaging was performed in order to identify the extent of the exposed V2 and obtain functional maps of V2 to determine array locations. Optical imaging procedures were similar to those described previously (*Lu et al., 2010*; *Li et al., 2013*).

Briefly, cortex was illuminated with 632 nm red light, and cortical reflectance was collected using Imager 3001 (Optical Imaging) while visual stimuli were presented. The imaging had a frame rate of 4 Hz and frame size of 540 × 654 pixels, corresponding to cortical region of 18 × 20 mm.

## Array recording

Array recordings were made 1 week after the implant surgery. On daily average, half of the channels had usable signals. Such condition usually lasted for 1.5–2 months, then the signal quality started to decline. Some arrays still had signals 6 months after the surgery. Parameters for the array are described in the surgery section. We used four 32 channels (4 × 8) for four hemispheres. In the 128 channels, 32 were V1 channels, in which 24 had neurons identified. In 96 V2 channels, 85 had neurons identified.

The electrophysiological recording system we used was AlphaLab SnR (Alpha Omega) 64-channel system. Neural signals were sampled at 22 kHz and filtered with a 800–7500 Hz bandpass filter. Daily recordings include mapping the RF locations, basic stimuli in fixation tasks for basic properties in each channel, and cell-specific stimuli in orientation-discrimination tasks.

For daily recordings, channel signals were spike-sorted (*Chen et al., 2014*) both online and off-line. For spike sorting, we first extracted discriminative features from spike waveforms based on principal component analysis (PCA) analysis, and then clustered these features by an expectation-maximization algorithm based on mixtures of multivariate t-distributions (*Shoham et al., 2003*). Signal-to-noise ratio was defined as the average signal size (peak-valley amplitude) of the action potential (signal) divided by the standard error of the residuals of the waveform (noise). The sorted response clusters were usually MU responses, but no longer separable based on their waveforms. Only clusters having a signal-to-noise ratio larger than 3.5 were considered, among which the highest signal-to-noise ratio cluster was considered the 'neuron' for that channel (other clusters were discarded). This neuron was either a SU (interspike intervals larger than 2.5 ms) or a MU. We obtained 723 neurons with this method and called it 'all cell' dataset. In this dataset, 70.7% (511/723) units were SUs and the rest (29.3%, 212/723) were MUs.

Similar to other array recording studies (i.e., *Ponce et al., 2017*), neurons recorded from the same channel over different days usually had different waveforms and/or tuning properties, thus were different cells. In addition to the full dataset (n = 723), we also analyzed data based on two stricter cell selection methods. In 'unique-unit' method, neurons were selected based on the 'all cell' dataset. We excluded potential duplicated units (i.e., had similar waveforms and tunings) recorded from the same electrodes on different days so that the remaining units were 'unique' ones. This means that the neurons in this dataset were either from different electrodes or from the same electrode but had different waveforms or tunings. For each electrode, we did this comparison on SUs first. Only when SUs were not available did we turn to compare and select MUs as supplements. With this method, we identified 287 neurons, in which 87.5% (251/287) were SUs and 12.5% (36/287) were MUs. In the third method, which was the strictest one, we identified only one cell for each channel (instead of averaging all the cells in that channel) based on the 'all cell' dataset. For 85 channels, we obtained 85 neurons, in which 73% (62/85) were SUs and 27% (23/85) were MUs. We found that the main results obtained using these three datasets were consistent.

## Quantification and statistical analysis

### Imaging data analysis

Functional maps (e.g., *Figure 1B*) were calculated based on t-tests (*Li et al., 2013*). For pixels in a t-map, its t-values were based on the pixel's response to two stimulus conditions (e.g., left eye vs. right eye). Polar maps (e.g., *Figure 1C, D*) were vector-summed single-condition t-maps (*Li et al., 2013*). Briefly, first we calculated single-condition t-maps comparing each stimulus condition with gray-screen blank condition, then these t-maps from all of the conditions were vector-summed to obtain a polar map (*Bosking et al., 1997*). For pixels in the polar maps, the color represented the preferred orientation or direction and the brightness represented the strength of the preference.

## Determining V1 and V2 electrodes

A picture of the cortex was obtained after the array implant (*Figure 1—figure supplement 2A–D*) and aligned to the imaging blood vessel map based on the surface blood vessel patterns (MATLAB,

projective transformation). Then the array locations were transformed to the ocular dominance map (*Figure 1—figure supplement 2E–H*), on which V1/V2 borders were clearly identified. From these maps, whether an electrode was in V1 or V2 was visually determined. For the four arrays shown in *Figure 1—figure supplement 2E–H*, the numbers of V1 channels were 8, 22, 0, and 2, respectively.

## Electrophysiology data analysis

For all neurons isolated in the spike sorting procedure, we performed three tests: orientation tuning, direction tuning, and MB orientation tuning. A neuron was used for the subsequent analysis only if it passed at least one of these three tests. According to these criteria, we obtained 723 neurons, of which 677 passed orientation test, 78 passed direction test, and 70 passed MB orientation test. There were nine neurons showing strong direction bias (DSI > 0.5) to grating stimuli but did not show direction bias in direction test with RDs. We excluded these neurons in the subsequent analysis as a precaution that they may behave abnormally in MB stimuli. In V1, we obtained 97 neurons, of which 93 passed the orientation test, 3 passed the direction test, and 2 passed the MB test. The neuron population is illustrated in *Figure 1—figure supplement 1F*.

## Orientation tuning and direction tuning

Unless otherwise specified, neuron's responses to a particular stimulus were measured as its baseline-subtracted mean spike rate during the stimulus presentation period. The baseline activity was the mean spike rate during the 200 ms period immediately before the stimulus onset.

Orientation tuning was calculated based on neuron's responses to sine-wave gratings. Direction tuning was calculated from neuron's responses to moving RDs. Response functions were fitted with a modified von Mises function (*Mardia, 1972*; *Li et al., 2013*):
$y = a + b1*e^{c1*\cos(x-d1)} + b2*e^{c2*\cos(x-d2)}$, in which x represents the directions tested, y represents the corresponding firing rates and is a function of x, where a is the baseline offset, and (b1, b2), (c1, c2), (d1, d2) determine the amplitude, shape, and position of the tuning curve, respectively. The OSI and DSI were calculated as described previously (*Li et al., 2013*; *Hu et al., 2018*): OSI = 1 null-orientation response/preferred orientation response; DSI = 1 anti-preferred direction response/preferred direction response.

Criteria for orientation-selective and direction-selective neurons were that their OSI or DSI was larger than 0.5, respectively, and the goodness of fit should be larger than 0.7.

## RF analysis

For the RF data from the grid-like RF mapping, we fitted the responses with a 2-D Gaussian function and only those having a goodness of fit larger than 0.7 were further analyzed. The center of the fitting function was recorded as the center of the RF. The RF size was calculated as 2 × 1.96 × SE-0.8, in which SE was the standard error of the Gaussian and 0.8 was the size of the grating stimuli.

For the RF data measured with bars, one-dimensional Gaussian was used for horizontal and vertical mappings, respectively. The RF center was determined by the Gaussian centers in x and y dimensions, and the sizes on both dimensions were estimated the same way as described above.

For most neurons, the RF information was obtained from the 'grid' mapping data. The 'bar' mapping data was used only when the 'grid' data was not good enough to meet the fitting criteria.

## Orientation tuning to MB/TB

Orientation tuning to MB stimuli was calculated similar to the orientation tuning to luminance gratings (described above). Tunings to MB1 and MB2 were calculated separately. The criteria for a MB neuron were goodness of fit was larger than 0.7, both OSIs for MB stimuli were larger than 0.5, and the difference between the two preferred MB orientation was smaller than 30°. Four MB neurons did not show orientation selectivity to gratings (group E in *Figure 1—figure supplement 1F*). We also performed the same analysis for TB orientation; however, none of the recorded neurons exhibited orientation tuning to TB stimuli.

### Neurometric function

Psychometric functions (e.g., *Figure 2B*) were fitted with a Weibull function using the Psignifit toolbox (*Wichmann and Hill, 2001*). The behavioral threshold was obtained from the fitting function, where the correct rate was 75%.

For neuronal data, spike rate was measured from 100 to 700 ms after the stimulus onset and the baseline was subtracted. For each level of stimulus strength, a ROC was calculated based on the response distributions for the two orthogonal orientations (e.g., *Figure 2D*). A neurometric function (e.g., *Figure 2E*), which describes the neuron's orientation sensitivity, was calculated as the area under the ROC curves, plotted as a function of orientation signal. These neurometric functions were analyzed the same way as the psychometric functions described above.

### Choice probability

CP was used to evaluate the relationship between neural responses and behavioral choice (*Britten et al., 1996*). CP was also calculated using ROC analysis. For each level of stimulus strength, all trials (preferred and null orientations pooled) were separated into two groups based on the monkey's choice (preferred orientation target vs. null orientation target). In order to do this calculation, the monkey needed to make at least five choices for each target and each orientation. A ROC curve was then calculated from the two response distributions. The CP value for a neuron was the average of CPs from different levels of stimulus strengths (*Liu and Pack, 2017*; *Kang and Maunsell, 2012*). A CP was considered significant if it is located outside the 95% range in the distribution generated with bootstrap.

### PSTH and normalization

PSTHs (*Figure 2C, Figure 5*) were calculated based on the averaged spike rate in 1 ms bins, from which the baseline was subtracted. The baseline was defined as the mean spike rate during the 300 ms (200 ms for the DS surround modulation data) period before the stimulus onset. The PSTH was smoothed using a Gaussian (sd: 10 ms). PSTHs were normalized based on their peak values for comparison.

### d' analysis

d' was used to evaluate the response differences to two different stimuli. It was calculated as

$$\mathrm{d}' = (\mathrm{u}1 - \mathrm{u}2) / \sqrt{\left(\frac{\sigma_1^2 + \sigma_2^2}{2}\right)}$$

where $u_1, u_2$ are mean responses to two stimuli, and $\sigma_1, \sigma_2$ are the corresponding standard errors. A d' was considered significant if it was larger or smaller than the 95% population that was obtained in a bootstrap analysis.

### Response delay

Average PSTH was analyzed with a 10 ms sliding window and 2 ms step size. To estimate the visual response delay, we calculated d' by comparing the responses in each time window during 0–500 ms period with the pre-stimulus baseline (average of −300 to 0 ms for saccade tasks, −200 to 0 ms for fixation tasks). If three consecutive d's were larger than 0, then the mid-window time of the first window was recorded as the visual response delay (*Chen et al., 2017*). We had also used t-test to replace the d' evaluation; the results were the same.

In calculating other response times (e.g., MB orientation selectivity, surround suppression, and surround modulation, see *Figure 5*), the calculation was the same as above. Instead of comparing responses with baseline, responses to two different stimuli were compared. The results are presented in *Figure 5—figure supplement 1C, D*.

### Population response delay

Population response delays (*Figure 5*) were calculated using the method described in *Chen et al., 2017*. (1) For each neuron, a 10 ms sliding window (step size 2 ms) was used to compare the spike rate in the window with the pre-stimulus baseline. A d' was calculated, and its significance with 0 was evaluated with a bootstrap method. (2) In the neuron population, we calculated the proportion

of neurons having d's significantly larger than 0 at each time point. (3) We calculate a 'baseline proportion' and its standard error by averaging all the proportions in pre-stimulus time windows. (4) If there were three consecutive proportions in the post-stimulus window larger than 'baseline proportion' plus three standard error, then the mid-window time for the first window was recorded as the population response delay. Similarly, we calculated the time the two responses started to show difference.

## Cross-correlogram

CCG was used to evaluate the probability of coactivation of two neurons (*Figure 4*). For raw CCG, we used 1 ms window, and the function was the same as used by *Kohn and Smith, 2005*, *Smith and Kohn, 2008*, and *Chen et al., 2014*:

$$CCG(\tau) = \frac{\sum_{i=1}^{m} \sum_{\tau=1}^{n-\tau} c_1^i(t) c_2^i(t+\tau)}{\sqrt{\sum_{i=1}^{m} \sum_{\tau=1}^{n-\tau} c_1^i(t) \sum_{i=1}^{m} \sum_{\tau=1}^{n-\tau} c_2^i(t+\tau)}}$$

where $c_1, c_1$ represent spike trains of two neurons, $m$ represents the number of trials, $n$ represents the window number of each trial, and $\tau$ represents the time delay.

In order to remove the effects due to coactivation of the stimulus timing and firing rates, we calculated the CCG of trial shuffled responses. The final CCG (shuffle-corrected CCG) was calculated by subtracting the average of 1000 shift predictors from the raw CCG. All the CCGs presented in the results were shuffle-corrected CCG.

## Accumulative cross-correlograms

To quantify and compare different CCGs (*Figure 4D*), accumulative CCG (auCCG) was calculated by integrating the CCG curve between −40 and 40 ms (i.e., calculating the area size under the CCG curve).

## Acknowledgements

This work was supported by the National Natural Science Foundation of China (31625012, 31530029, 31371111) to HD Lu. We thank lab members J Lu, Y Xiao, SD Zhu, PC Li, C Han, M Chen, HR Xu, Y Fang, JY Wang, R Zhang, C Liang, Y Li, C Fang, K Yan, RD Tang, JT Xu, and WH Zhao for their valuable technical assistance and discussion.

## Additional information

### Funding

| Funder | Grant reference number | Author |
|---|---|---|
| National Natural Science Foundation of China | 31625012 | Haidong D Lu |
| National Natural Science Foundation of China | 31530029 | Haidong D Lu |
| National Natural Science Foundation of China | 31371111 | Haidong D Lu |

The funders had no role in study design, data collection and interpretation, or the decision to submit the work for publication.

### Author contributions

Heng Ma, Conceptualization, Data curation, Software, Formal analysis, Validation, Investigation, Visualization, Writing - original draft, Writing - review and editing; Pengcheng Li, Formal analysis, Investigation; Jiaming Hu, Software, Investigation; Xingya Cai, Qianling Song, Investigation; Haidong D Lu, Conceptualization, Resources, Data curation, Software, Supervision, Funding acquisition, Validation, Visualization, Writing - original draft, Project administration, Writing - review and editing

## Author ORCIDs

Heng Ma (iD) https://orcid.org/0000-0002-0322-278X
Jiaming Hu (iD) http://orcid.org/0000-0002-5306-445X
Xingya Cai (iD) http://orcid.org/0000-0001-7829-3833
Qianling Song (iD) http://orcid.org/0000-0001-9177-7429
Haidong D Lu (iD) https://orcid.org/0000-0003-1739-9508

## Ethics

Animal experimentation: Four hemispheres from two adult male macaque monkeys (*Macaca mulatta*) were used in this study. All procedures were performed in accordance with the National Institutes of Health Guidelines and were approved by the Institutional Animal Care and Use Committee of the Beijing Normal University (protocol number: IACUC(BNU)-NKCNL2013-13).

## Decision letter and Author response

Decision letter https://doi.org/10.7554/eLife.61317.sa1
Author response https://doi.org/10.7554/eLife.61317.sa2

# Additional files

## Supplementary files

• Supplementary file 1. Number of V2 electrodes (and neurons) from which motion boundary neurons were recorded.

• Supplementary file 2. Number of V2 electrodes (and neurons) from which choice probability was calculated. Note: The right hemisphere array in monkey W had a low electrode number since two thirds of its electrodes were located in V1.

• Transparent reporting form

## Data availability

Data and codes are available in the Mendeley dataset: https://doi.org/10.17632/fjy37kc8pd.3.

The following dataset was generated:

| Author(s) | Year | Dataset title | Dataset URL | Database and Identifier |
|---|---|---|---|---|
| Ma H, Li P, Hu J, Cai X, Song Q, Lu HD | 2021 | Processing of motion-boundary orientation in macaque V2 | http://dx.doi.org/10.17632/fjy37kc8pd.3 | Mendeley Data, 10.17632/fjy37kc8pd.3 |

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
