## [Decision Letter]

**Acceptance summary:**

The study by Lu and colleagues is an in-depth investigation into the selectivity of V2 neurons in the primate for motion-boundaries (MB), the underlying functional circuitry and the potential perceptual contribution of these neurons. Previous studies in anesthetized monkeys demonstrated that V2 contains neurons selective for the orientation of MB, and the authors previously showed that these neurons form a map of MB orientation. In this study, they confirm these previous findings in awake animals and further demonstrate that the responses of MB neurons correlate with the animals' behavioral performance in a MB orientation-discrimination task. The activity of MB neurons also correlates with that of direction selective V2 neurons. These findings are important for our understanding of the functional circuitry underlying shape perception in primates.

**Decision letter after peer review:**

Thank you for submitting your article "Processing of motion-boundary orientation in macaque V2" for consideration by *eLife*. Your article has been reviewed by 2 peer reviewers, one of whom is a member of our Board of Reviewing Editors, and the evaluation has been overseen by Timothy Behrens as the Senior Editor. The following individual involved in review of your submission has agreed to reveal their identity: Alessandra Angelucci (Reviewer #2).

The reviewers have discussed the reviews with one another and the Reviewing Editor has drafted this decision to help you prepare a revised submission.

As the editors have judged that your manuscript is of interest, but as described below that additional revisions and data are required before it is published, we would like to draw your attention to changes in our revision policy that we have made in response to COVID-19 (https://elifesciences.org/articles/57162). First, because many researchers have temporarily lost access to the labs, we will give authors as much time as they need to submit revised manuscripts. We are also offering, if you choose, to post the manuscript to bioRxiv (if it is not already there) along with this decision letter and a formal designation that the manuscript is "in revision at *eLife*". Please let us know if you would like to pursue this option. (If your work is more suitable for medRxiv, you will need to post the preprint yourself, as the mechanisms for us to do so are still in development.)

Summary:

The study "Processing of motion-boundary orientation in macaque V2" by Lu and colleagues is an in-depth investigation into the selectivity of V2 neurons in the primate for motion-boundaries (MB), the underlying functional circuitry and the potential perceptual contribution of these neurons. Previous studies in anesthetized monkeys demonstrated that V2 contains neurons selective for the orientation of MB, and the authors previously showed that these neurons form a map of MB orientation. In this study, the authors confirm these previous findings in awake animals and further demonstrate that the responses of MB neurons correlate with the animal's behavioral performance in a MB orientation-discrimination task. Moreover, the authors demonstrate that the responses of direction selective (DS) V2 neurons are correlated in time with those of MB neurons and that the visual response latencies of DS cells are consistent with a model in which MB neurons integrate motion signals from DS neurons in V2 itself to extract information about the orientation of the MB. The results are novel and important, the study appears overall well executed.

However the presentation of the manuscript needs improvement by addition of more experimental details, explanations of rationales for some of the analyses, and substantial proof reading to clarify the language. Despite a large numbers (n) of neurons from array recordings, interpretation is in parts difficult because of the low n for neurons showing MB selectivity. To understand the robustness of MB results better, more detail is required in the main manuscript about the distribution of results for MB neurons across array electrodes, across the two animals as well as the relationship of MB results to stimulus size and position with regards to the neuronal selectivity. Finally, the interpretation of the cross-correlation and latency studies is somewhat weak and needs additional analysis and justification.

Detailed comments are provided below.

Essential revisions:

1. To understand the robustness of the results obtained for MB selective neurons better, more detail is required in the main manuscript about the distribution of MB neuron results across array electrodes, across the two animals and about the relationship of stimulus size and position to the neuronal selectivity:

– Figure 1: to what extent would the results be influenced by surround inhibition? Show RF positions and sizes of recorded neurons on the same plot as the stimulus position and size used.

– p6, line 140: 10.9% (70/642) of V2 neurons were MB neurons. From how many individual electrode points did they come in each of the monkeys and hemispheres?

– p8, line 207-226 and Figure 2. CP and neurometric measurements

Given the low n, how many unique electrode points in each of the two monkeys do the data stem from?

– p9: V1/V2 comparison first paragraph ("significantly higher percentage MD detection capability"). Given the small sample size in V1, what is the statistics on differences in distribution – is this really significant given the small sample size ? Also, where "could not be fitted" or "is well fitted" is mentioned, please provide the specific fit, goodness-of-fit measure and statistical criterion in the text.

– p9: V1/V2 comparison. Figure 3. Please show graphically V1 RF positions and size in relation to stimulus position and size. Were the same stimulus parameters used for V1 as for V2. If yes, could surround suppression account for the poorer results in V1 (MB tuning distribution, lower OSI, lower CP)?

– p11. Figure 4: Could the coverage of the DS RF by the preferred motion direction component of the MB stimulus explain some of the interneuronal correlation? For example, between pref and null orientation, when the angle of the dividing line changes, the RF parts that are exposed to preferred and null motion change around the line. Please mention in the main manuscript the range of cortical distances over which pairs of neurons have been simultaneously recorded.

– p12, lines 320- 328: Were TBs resented at pref or null motion direction or both. Were monkeys rewarded for correct choices only as for MBs?

– p14, line 388: please give n of neurons for time course data. Also state for delay times whether this is the mean (across how many neurons?) and what the standard deviation is.

– The authors should discuss how these results might relate to the distribution across different types of stripes (thick, thin, interstripe) across their recording sites in V2.

2. The authors used optical imaging to position the recording arrays to MB-selective domains in V2. How was the exact position of the recording channels relative to the imaging maps of V1 and V2 recovered? Using imaging as the only guidance is relatively imprecise. Could additional criteria to discern between V1 and V2 cells, such as receptive field (RF) size and retinotopy be used?

The authors state that spike sorting was used to isolate single units (SUs) from multiunits (MUA). On p. 4, lines 91-95 they state: "…we found that neurons recorded from the same channel over different days usually had different waveforms and/or tuning-properties, thus were different cells. We compared the results obtained either using the whole dataset ("all-cells", n=723) or the "one-cell-per-channel" dataset (n=85). However, neither approach is accurate, as the first approach will count some of the same neurons twice, and the second approach averages across different, rather than the same, cells. The authors should spike sort the cells, then select SUs based on the spike waveform and tuning across days. That should be their n.

3. Figure 2F: Please describe in the figure legend what the black curve and red dots are. Is this the mean of all neuronal neurometric functions? Also please add the psycometric function on this plot for reference.

4. Lines 223-225: "In 17 neurons recorded in one array, we also tested neural-behavioral relevance with different levels of dot brightness. The results were similar as those from coherence stimuli (Figure S2B-E)". In fact, the results are not so similar. It seems much fewer neurons had CPs that were significantly larger than chance. How do the authors interpret this result?

5. Lines 267-276. Wouldn't a better comparison be between the CP distribution of MB neurons vs non-MB neurons, instead of vs. all V2 neurons? The same apply to the comparison with V1, although We do realize that only 2 cells in their sample passed the test for MB neuron classification.

6. Lines 303-313: a good control here would be to look at the correlation with DS cells whose preference does not match the motion of the random dots generating the MB.

7. Analysis of time correlations. It is unclear to us why the authors interpret CCGs between the responses of DS and MB neurons peaking at zero, as supporting their model of MB responses resulting from integration of DS responses. Doesn't such a model predict that the responses of DS cells should rather precede those of MB cells?

8. Analysis of response time course. The authors find that the latency of MB selectivity falls within the range of latencies of surround suppression in DS neurons, and from this finding conclude that this suggests that V2 DS neurons contribute to the generation of MB responses in V2 MB neurons. The rationale for why surround suppression in DS neurons contributes to MB is unclear to us. MB responses occur within the RF of the V2 MB cells, so how does surround suppression play a role in their generation? Perhaps, we are failing to understand something here, but this needs to be better explained. Also, given the hypothesis here is that DS neurons contribute to MB responses, shouldn't the authors look at the onset latency of direction selectivity in DS neurons? Our rationale is that inputs regarding the direction of RDs need to be fed to the MB neurons in order to extract the MB orientation. The authors need to provide a better explanation of what model they have in mind for how DS cells contribute to the extraction of MB in MB neurons.

9. Lines 440-442. A simple model to support the statement that MB detection results from population coding would help here. At a minimum the authors should provide a sense of how this could be achieved. Also, why couldn't MB detection result from the activity of the fewer neurons that show neurometric functions and thresholds similar to the psychometric functions?

10. Was eye movements correction applied to the data analysis? how? Please specify.

11. Line 825: what algorithm/method was used for spike sorting? Please describe in the Methods.

12. Figure S1 legend line 988-989: the definition of MB neurons provided here does not match that provided in the results, according to which MB neurons were considered those that respond to MB AND are orientation selective. So group E in panel E does not fit this definition.

13. In the methods, the authors often refer to their previous publications for method and analysis details. However, at brief description should be provided in this manuscript. For example, on line 867: how were OSI and DSI defined?

14. Monkey task (page 6, line 162 and methods line 778): to me, the expression that the monkeys did discriminated between an acute or obtuse angle does not mean much in this contexts I am unclear which angle they refer to. If the authors meant, the monkeys discriminated whether the MB line was tilted e.g. left or right of vertical (or another axis), they need to say so more explicitly. The specific tasks the monkey had to carry out need a better description and figures to understand what stimuli configurations would lead to which choices.

[Editors' note: further revisions were suggested prior to acceptance, as described below.]

Thank you for resubmitting your work entitled "Processing of motion-boundary orientation in macaque V2" for further consideration by *eLife*. Your revised article has been reviewed by 2 peer reviewers, one of whom is a member of our Board of Reviewing Editors, and the evaluation has been overseen by Timothy Behrens as the Senior Editor.

We found that you have thoroughly revised the manuscript and performed additional analyses according to the Reviewer's comments, which is appreciated. The manuscript has been improved but there are some remaining issues that need to be addressed, as outlined below:

1. We are somewhat concerned about the lack of significant difference in the receptive field (RF) size of V1 and V2 neurons. The reported 2.5 deg mean RF size for V1 cells is certainly inconsistent with what has been reported in the literature for parafoveal V1 neurons (more around 0.8-1 deg). This discrepancy to the published literature should be discussed in the paper – alongside the potential reasons for this, specifically:

a. The most likely cause for this is eye movements (in fact the fixation window is reported to be 1.2deg, which is rather coarse for V1). The question here is in what way this imprecision in the mapping of V1 RFs may have affected the results, e.g. the reported differences between V1 and V2 cells. You should at the very least add a discussion of this issue.

b. The illustrated placement of the arrays looks like that for 3 out of 4 arrays very few electrodes would be safely in V1, which could also explain the lack of a difference in RF size. You should specify in the methods of the paper how exactly they determined functionally whether each electrode was in V1 or V2.

c. The V1 RF measurements shown in the authors' response in comparison with V2 should also be included in the supplementary figure (could be added to the supplement to Figure 3).

2. Authors' Reply point 7). Here, you acknowledge that the zero lag in the CCG between DS and MB neuron responses is inconsistent with your hypothesis of DS inputs contributing to the MB responses. Yet, in the Results section of the paper you do not say so, rather you still maintain that the results are consistent with your hypothesis but concede that alternative/additional mechanisms may also be at play. If you indeed agree this result is inconsistent with your hypothesis, you should state so. Alternatively, you should clarify in the manuscript why you think this result may still be consistent with your hypothesis.

3. Authors' Reply point 9. The two alternative model schemes do a good job at clarifying why and you think that surround suppression may play a role in the generation of MB responses and how. We recommend to include this figure in the discussion of the paper.

4. Authors' Reply point 11. Please include in the manuscript your explanation provided here of why the 4 neurons in group E were included in the MB population analysis. This is ok, but this needs to be clear in the manuscript (methods and figure legend need to be consistent).

5. line 1041-1051: ".… sorted response clusters were usually multi-unit responses,.…" "Neurons with inter-spike-intervals larger than 2.5 ms were identified as SUs." The paragraph reads as if most data presented in the paper were multi-unit, but were re-classified as SU based on spike timing only, which by itself is not enough. The main results (line 106) however state that SUs were identified on unique spike wave form and asserts that most cells are SU. We presume this means that the SUs were clearly separable from the remaining MU or noise.

There need to be clear, consistent statements in both Results and Methods, as the reader needs to know (i) whether they are dealing with SU or MU results and (ii) if both were combined whether specific results about basic MB tuning differ for clear SUs und MUs.

6. The reviewers would be grateful if you could please always point to the pages in the revised manuscript where the revisions were made. It really saves reviewers a lot of time.

---

## [Author Response]

Essential revisions:1. To understand the robustness of the results obtained for MB selective neurons better, more detail is required in the main manuscript about the distribution of MB neuron results across array electrodes, across the two animals and about the relationship of stimulus size and position to the neuronal selectivity:– Figure 1: to what extent would the results be influenced by surround inhibition? Show RF positions and sizes of recorded neurons on the same plot as the stimulus position and size used.

Figure 3—figure supplement 1 shows the spatial relationship between V2 RFs and the stimuli (A-D). The dotted circles represent the 4° stimulus patch. Since the stimuli were not adjusted for individual neurons, their size was chosen to slightly larger than the V2 RF (2.47±0.85°), Therefore, there should be some surround inhibition for these neurons. However, this effect should be similar for the MB and non-MB neurons, since the sizes of their RFs were similar (C, same as in Figure 1J), and their RF centers were located in similar distances to the stimulus center (D). In addition, the major stimuli we used were oppositely moving random dots, which were different from those homogenous stimuli that usually cause strong surround inhibition.

Similarly, V1 RFs are also plotted in Figure 3—figure supplement 1E-H, and related to the answers to another question below.

– p6, line 140: 10.9% (70/642) of V2 neurons were MB neurons. From how many individual electrode points did they come in each of the monkeys and hemispheres?

The 70 MB neurons were from 26 individual electrodes. In the revised manuscript, we added a table showing the detailed information (Supplementary File 1).

– p8, line 207-226 and Figure 2. CP and neurometric measurementsGiven the low n, how many unique electrode points in each of the two monkeys do the data stem from?

In the revised manuscript, we included a table (Supplementary File 2) showing the number of unique electrodes from each hemisphere that neurons’ CPs were calculated.

– p9: V1/V2 comparison first paragraph ("significantly higher percentage MD detection capability"). Given the small sample size in V1, what is the statistics on differences in distribution – is this really significant given the small sample size ? Also, where "could not be fitted" or "is well fitted" is mentioned, please provide the specific fit, goodness-of-fit measure and statistical criterion in the text.

In the revised manuscript, we added statistics to support the above conclusions. Author response table 1 shows the neuron numbers being compared (same as in Figure 3A). V1 and V2 were different in their cell distributions in the 3 groups (Chi-squared test, χ²=14.2，p=8.25×10^-4^). In addition, the mean orientation index (OSI) for MB stimuli was larger for V2 neurons than for V1 neurons (calculated from the neurons in the first two rows whose responses could be well fitted, i.e. R²≥0.7). We also added fitting criterion into the main text: The fitting criteria for MB was the same as for gratings, i.e. with von Mises function, R²<0.7 was considered not fitted; R²≥0.7 was well fitted.

**Author response table 1. resptable1:** 

	V1	V2
MB neurons	2	70
Well fitted (R²≥0.7), but OSI<0.5 or two preferred MB orientation were different (>30°);	15	169
Can not fit (R²<0.7)	76	403

– p9: V1/V2 comparison. Figure 3. Please show graphically V1 RF positions and size in relation to stimulus position and size. Were the same stimulus parameters used for V1 as for V2. If yes, could surround suppression account for the poorer results in V1 (MB tuning distribution, lower OSI, lower CP)?

The same stimuli were used for V1 and V2 neurons in our experiments. In the figure for question 1 above, we also plotted V1 RF positions and sizes. V1 and V2 neurons we recorded had similar RF sizes (V1: 2.50±1.18°; V2: 2.47±0.85°; p=0.77, Wilcoxon test). The distances between the RF centers and stimulus center were slightly larger for V1 neurons (V1: 0.47±0.41°; V2: 0.33±0.24°; p=2.97×10^-4^, Wilcoxon test), but the difference was relatively small (0.14°). We also compared the firing rate of V1 and V2 neurons in two arrays having sufficient numbers of V1 neurons, and observed no differences (Monkey S, right hemisphere: V2 13.95±8.60 spike/s; V1: 12.63±6.84 spike/s, p=0.45, t-test; Monkey W，right hemisphere: V2: 15.55±6.34 spike/s; V1: 15.36±13.60 spike/s, p=0.93, t-test). Thus, we believe that the surround inhibition caused by the common stimuli was similar for V1 and V2, and was not the cause for the poorer results in V1.

– p11. Figure 4: Could the coverage of the DS RF by the preferred motion direction component of the MB stimulus explain some of the interneuronal correlation? For example, between pref and null orientation, when the angle of the dividing line changes, the RF parts that are exposed to preferred and null motion change around the line. Please mention in the main manuscript the range of cortical distances over which pairs of neurons have been simultaneously recorded.

The reviewer is correct that a DS neuron might have different response levels to the two MB orientations (preferred and null) due to the change of preferred motion in its RF. However, since this factor was random for the two MB orientations, it would not cause a biased enhancement of the MB-DS correlation for a particular MB orientation. To confirm this, we identified and removed DS neurons (n=4) that had large RF coverage differences in preferred and null conditions and found that the auCCG was still significantly larger for the preferred condition (preMB: 0.10±0.016 vs. nullMB: 0.05±0.011, p=3.11X10^-4^, t-test). The cortical distances of paired neurons in Figure 4A-C had a range of 0.93~1.1 mm. We have added this information into the main text.

– p12, lines 320- 328: Were TBs resented at pref or null motion direction or both. Were monkeys rewarded for correct choices only as for MBs?

Yes, the TB stimuli were presented at both preferred and null directions. The monkeys were rewarded for correct choices. However, the correct rate for TB was around 50%.

– p14, line 388: please give n of neurons for time course data. Also state for delay times whether this is the mean (across how many neurons?) and what the standard deviation is.

We have added n=9 for the number of DS neurons. We calculated delay times with two methods, the one described in the main text was based on population time courses (no standard deviation), and another based on individual neurons’ time courses (Figure 5—figure supplement 1D). We’ve now described this in the main text.

– The authors should discuss how these results might relate to the distribution across different types of stripes (thick, thin, interstripe) across their recording sites in V2.

We have added discussion about the stripe types in the manuscript (line 531-542). Since CO histology is unavailable for these monkeys. The stripe types were estimated based on optical imaging maps (ocular dominance, orientation, color, motion direction). To separate thick and pale stripes, we use a width ratio previously described (thick : pale=2 : 1.5, Shipp and Zeki 1989). Figure 1—figure supplement 2 shows the locations of the 4 arrays overlaid on the corresponding color vs. luminance maps. The white dotted lines represent V1/V2 borders. The thin stripes were identified based on the color/luminance patches in V2 (black and white patched in V2). Thick (between two red dashed lines) and pale (between green and red dashed lines) stripes were identified between the thin stripes and according to their width ratio. For all V2 channels, the total numbers of channels located in thin, thick and pale stripes were: 15, 44, and 37, respectively. Figure 1—figure supplement 2M lists neuron numbers according to their stripe and tuning types. Note that the tuning types were not mutually exclusive (e.g. a MB neuron could also be an orientation neuron). The percentage values were calculated for each stripe types. Although a statistical analysis was not available due to the overlapped tuning types, one can still draw some conclusions from the table. For example, DS neurons were more likely found in thick stripes, while MB neurons were more likely found in thick and pale stripes. These results were consistent with previous findings (Hubel and Livingstone, 1987; Levitt et al., 1994; Shipp and Zeki, 2002; Lu et al. 2010; Chen et al. 2016).

2. The authors used optical imaging to position the recording arrays to MB-selective domains in V2. How was the exact position of the recording channels relative to the imaging maps of V1 and V2 recovered? Using imaging as the only guidance is relatively imprecise. Could additional criteria to discern between V1 and V2 cells, such as receptive field (RF) size and retinotopy be used?

We estimated the electrode locations by comparing the blood vessel pictures taken before and after the array implants. We agree with the reviewer that this method is relatively imprecise (the maximum offset we estimate is within 200 um). According to the reviewers’ suggestion, we compared the RF sizes and retinotopy for the V1 and V2 neurons. However, no significant trends were observed, and thus did not help in determining the V1/V2 borders. In one array we also tested the monocular/binocular driven properties of the neurons. Probably due to the superficial layers the electrodes located (layer 2/3), neither did we observe monocularly-driven neurons even in electrodes obviously located in V1. A post-mortem histology would help but is not available at this time.

The authors state that spike sorting was used to isolate single units (SUs) from multiunits (MUA). On p. 4, lines 91-95 they state: "…we found that neurons recorded from the same channel over different days usually had different waveforms and/or tuning-properties, thus were different cells. We compared the results obtained either using the whole dataset ("all-cells", n=723) or the "one-cell-per-channel" dataset (n=85). However, neither approach is accurate, as the first approach will count some of the same neurons twice, and the second approach averages across different, rather than the same, cells. The authors should spike sort the cells, then select SUs based on the spike waveform and tuning across days. That should be their n.

Following the reviewers’ suggestions, we used a “unique-unit method”. In unique-unit method, we first spike sorted the neurons according to their waveforms, then identified single-units (SUs) that had unique waveforms or tuning properties. If for a channel no SU was found from all the recording days, we used multi-units (MUs) from different cell types as supplements. We obtained 287 unique-units (251 SUs and 36 MUs) in V2 with this method. We repeated the data analysis on this cell set and obtained similar results (Figure 5—figure supplement 3) as those obtained with the cell sets described in the original manuscript. Thus, we have 3 cell identification methods: “all-cells n=723”, “unique-units, n=287”, and “one-cell-per-channel, n=85”, in order of the selection strictness. The main conclusions obtained from these 3 datasets were the same. We added this into the manuscript.

Additional notes: In “one-cell-per-channel” method, we identified only one cell for each channel (instead of averaging all the cells in that channel), and thus this method is the strictest one among the 3 methods.

3. Figure 2F: Please describe in the figure legend what the black curve and red dots are. Is this the mean of all neuronal neurometric functions? Also please add the psycometric function on this plot for reference.

In Figure 2 legend we have added appropriate descriptions. The red dots are mean values of the neurometric functions and the black curve is the fitting curve. In Figure 2F, we also added the psychometric function obtained by averaging the behavioral performance during the 15 neurons’ recordings (green dots). Similarly, average psychometric function was added into Figure 2—figure supplement 1C for neurons tested with brightness stimuli.

4. Lines 223-225: "In 17 neurons recorded in one array, we also tested neural-behavioral relevance with different levels of dot brightness. The results were similar as those from coherence stimuli (Figure S2B-E)". In fact, the results are not so similar. It seems much fewer neurons had CPs that were significantly larger than chance. How do the authors interpret this result?

We agree that the overall CP was lower in neurons tested with dot brightness. This might be due to the relatively easiness for the brightness task. In 5-6 brightness levels, the monkey achieved more than 80% correct rate. In contrast, the same monkey achieved 80% correct rate in only 3-4 levels in the coherence tasks. An easier task requires less effort and may lead to lower CP values. Nevertheless, in both tasks, the average CPs were both larger than 0.5 and portions of neurons had CP values larger than 0.5. We revised the manuscript to reflect both the differences and similarity.

5. Lines 267-276. Wouldn't a better comparison be between the CP distribution of MB neurons vs non-MB neurons, instead of vs. all V2 neurons? The same apply to the comparison with V1, although We do realize that only 2 cells in their sample passed the test for MB neuron classification.

We added a figure (Figure 3—figure supplement 2) into the manuscript, in which CP distributions for MB and non-MB neurons are compared. In V2, although non-MB neurons had a lower mean CP (0.53) than the MB neurons’ (0.56), this value was still significantly larger than 0.5 (t-test, p<0.001). There were 31.5% (34/108) non-MB neurons had a CP larger than 0.5 (bootstrap test, p<0.05). In comparison, this proportion in MB neurons was 46.9% (15/32). In V1, non-MB neurons had a non-significant mean CP (0.51) and the portion of significant CP neurons was low (21%). These results were now added into the manuscript.

6. Lines 303-313: a good control here would be to look at the correlation with DS cells whose preference does not match the motion of the random dots generating the MB.

We took the reviewers’ suggestion and analyzed DS-MB pairs in which DS neurons were not optimally stimulated. The results are shown in Author response image 1. It shows that the CCGs for preferred MB and null MB do not differ (5 pairs, t-test, p=0.26), which is consistent with our hypothesis. In the revised manuscript, we have included this results.

7. Analysis of time correlations. It is unclear to us why the authors interpret CCGs between the responses of DS and MB neurons peaking at zero, as supporting their model of MB responses resulting from integration of DS responses. Doesn't such a model predict that the responses of DS cells should rather precede those of MB cells?

Indeed, according to our model, the DS neurons made contribution to the MB neurons, and CCG should peak at the right side of zero. However, the peak we observed was centered at zero (Figure 3A). In addition, similar peaks were also observed in control conditions (e.g. real line in Figure 3B). These phenomena were not predicted by our model, and possibly due to other extra connections. For example, both DS and MB neurons might receive certain common inputs, or these neurons had bi-directional interactions (Bastos and Schoffelen 2016). Similar phenomena were also observed in previous pair-recording studies (e.g. orientation neurons in V1 and V2 in Roe and Ts’o 2015). We revised the interpretation in the manuscript accordingly.

8. Analysis of response time course. The authors find that the latency of MB selectivity falls within the range of latencies of surround suppression in DS neurons, and from this finding conclude that this suggests that V2 DS neurons contribute to the generation of MB responses in V2 MB neurons. The rationale for why surround suppression in DS neurons contributes to MB is unclear to us. MB responses occur within the RF of the V2 MB cells, so how does surround suppression play a role in their generation? Perhaps, we are failing to understand something here, but this needs to be better explained. Also, given the hypothesis here is that DS neurons contribute to MB responses, shouldn't the authors look at the onset latency of direction selectivity in DS neurons? Our rationale is that inputs regarding the direction of RDs need to be fed to the MB neurons in order to extract the MB orientation. The authors need to provide a better explanation of what model they have in mind for how DS cells contribute to the extraction of MB in MB neurons.

We took the reviewers’ suggestion and considered two DS-MB models, in which surround suppression plays different roles (see Figure 5—figure supplement 4).

In model 1, MB detection mainly relies on the center-surround mechanism of the DS neurons, which detect the motion contrast at the MB. Our previous work has shown that, in this condition DS neurons are activated optimally (Figure 5 in Hu et al. 2018). Also, the precise location of the MB is detected at the same time. In the primate visual system, such a center-surround mechanism appears to be used as a general strategy in detecting changes of first order cues at edges (e.g. luminance, color, disparity etc.).

In model 2, MB detection is achieved by comparing the activation of two DS neurons, and surround suppression plays a less significant role in this condition. We have shown that DS neurons are sub-optimally activated in such conditions (Figure 5 in Hu et al. 2018), and a precise MB location is unavailable.

Thus, we mainly considered Model 1 in our manuscript. The onset time of direction selectivity for DS neuron was 49 ms. It was faster than the MB-selectivity time of MB neurons (85 ms) and thus was not against Model 2. In the revised manuscript we added relevant discussions.

9. Lines 440-442. A simple model to support the statement that MB detection results from population coding would help here. At a minimum the authors should provide a sense of how this could be achieved. Also, why couldn't MB detection result from the activity of the fewer neurons that show neurometric functions and thresholds similar to the psychometric functions?

In our results, the MB orientation index is a continuous distribution (Figure 3A). Similarly, the CP distribution does not show a bimodal distribution (Figure 3B). These results indicate that MB information is likely coded in a distributed fashion in V2. Since single neuron’s response is intrinsically noisy, a detection mechanism only relies on a few high-performance neurons may not be the best solution. Population coding strategy overcomes this by using all available useful information. For example, a weighted summation model (Jazayeri and Movshon 2006) can be used in MB detection task. In this model, a likelihood function (i.e. a probability density function) is calculated based on the weighted-summing of all activated neurons (optimally and sub-optimally). Based on the likelihood function, a series of behavioral tasks (e.g. detection, discrimination) can be achieved. We have added this discussion into the manuscript.

10. Was eye movements correction applied to the data analysis? how? Please specify.

We continuously monitored the eye movements during the awake experiments. The fixation window was a 1.2° square. If the eye position moved outside this window for 20 ms, the task was aborted and the data was discarded. Therefore, all the data presented in the manuscript was collected for eye movement within ±0.6°. During the awake experiments, the measured eye position usually had some baseline drift over the time due to the system factors. We monitored the eye position and corrected the drift manually when necessary.

We calculated the mean eye movement for each monkey during the fixation period: Monkey S: 0.05° (horizontal) and 0.08° (vertical); Monkey W: 0.05° (horizontal) and 0.1° (vertical). These values were relatively small. We did not find a correlation between the eye movement size and the main results (behavioral performance, CP, and auCCG).

11. Line 825: what algorithm/method was used for spike sorting? Please describe in the Methods.

We added the spike sorting procedures into the Method.

12. Figure S1 legend line 988-989: the definition of MB neurons provided here does not match that provided in the results, according to which MB neurons were considered those that respond to MB AND are orientation selective. So group E in panel E does not fit this definition.

Our definition of MB neuron was: Their MB responses can be fitted by a von Mises function, and exhibit orientation selectivity to MB stimuli (i.e. OSI_MB_ >0.5). In Figure 1—figure supplement 1E, the 4 neurons in group E were not orientation neurons (did not reach the criteria for orientation selectivity to grating stimuli). However, they exhibited MB orientation selectivity. Thus, these 4 neurons were considered MB neurons and included in the MB results.

13. In the methods, the authors often refer to their previous publications for method and analysis details. However, at brief description should be provided in this manuscript. For example, on line 867: how were OSI and DSI defined?

We now added descriptions in methods and do not relies on previous publications.

OSI=1- null-orientation response /preferred orientation response;

DSI=1- anti-preferred direction response /preferred direction response;

14. Monkey task (page 6, line 162 and methods line778): to me, the expression that the monkeys did discriminated between an acute or obtuse angle does not mean much in this contexts I am unclear which angle they refer to. If the authors meant, the monkeys discriminated whether the MB line was tilted e.g. left or right of vertical (or another axis), they need to say so more explicitly. The specific tasks the monkey had to carry out need a better description and figures to understand what stimuli configurations would lead to which choices.

We revised the text and Figure to reflect the details of animal tasks.

References:

Anzai A, Peng X, Van Essen DC. Neurons in monkey visual area V2 encode combinations of orientations. Nat Neurosci. 2007;10:1313-1321.

Bastos, A. M., and Schoffelen, J.-M. (2016). A Tutorial Review of Functional Connectivity Analysis Methods and Their Interpretational Pitfalls. Frontiers in Systems Neuroscience, 9:175, 1–23.

Bredfeldt, C. E., and Cumming, B. G. (2006). A simple account of cyclopean edge responses in macaque V2. Journal of Neuroscience, 26(29), 7581–7596.

Britten, K. H., Shadlen, M. N., Newsome, W. T., and Movshon, J. a. (1992). The analysis of visual motion: a comparison of neuronal and psychophysical performance. The Journal of Neuroscience : The Official Journal of the Society for Neuroscience, 12(12), 4745–4765.

Chen, M., Li, P., Zhu, S., Han, C., Xu, H., Fang, Y., Hu, J., Roe, A.W., and Lu, H.D. (2016). An Orientation Map for Motion Boundaries in Macaque V2. Cereb. Cortex 26, 279–287.

DeYoe, E.A. and Van Essen, D.C. (1985). Segregation of efferent connections and receptive field properties in visual area 2 of the macaque. Nature 317, 58–61.

El-Shamayleh, Y., and Anthony Movshon, J. (2011). Neuronal responses to texture-defined form in macaque visual area V2. Journal of Neuroscience, 31(23), 8543–8555.

Gattass, R., Gross, C. G., and Sandell, J. H. (1981). Visual topography of V2 in the macaque. Journal of Comparative Neurology, 201(4), 519–539.

Hegdé, J., and Van Essen, D. C. (2007). A comparative study of shape representation in macaque visual areas V2 and V4. Cerebral Cortex, 17(5), 1100–1116.

Hu, J., Ma, H., Zhu, S.,, Li, P., Xu, H., Fang, Y., Chen, M., Han, C., Fang, C., Cai, X., Yan, K., Lu, H.D. (2018) Visual Motion Processing in Macaque V2. Cell Reports. 25, 157–167..

Hubel D. H., Livingstone M. S. (1987). Segregation of form, color and stereopsis in primate area 18. J. Neurosci. 7, 3378–3415

Ito, M., and Komatsu, H. (2004). Representation of Angles Embedded within Contour Stimuli in Area V2 of Macaque Monkeys. Journal of Neuroscience, 24(13), 3313–3324.

Jazayeri, M., and Movshon, J. A. (2006). Optimal representation of sensory information by neural populations. Nature Neuroscience, 9(5), 690–696.

Jiapeng Yin, Hongliang Gong, Xu An, Zheyuan Chen, Yiliang Lu, Ian M. Andolina, Niall McLoughlin, Wei Wang Breaking cover: neural responses to slow and fast camouflage-breaking motionProc Biol Sci. 2015 282: 20151182.

Kobatake, E., and Tanaka, K. (1994). Neuronal selectivities to complex object features in the ventral visual pathway of the macaque cerebral cortex. Journal of Neurophysiology, 71, 856–867.

Levitt, J. B., Kiper, D. C., and Movshon, J. A. (1994). Receptive fields and functional architecture of macaque V2. Journal of Neurophysiology, 71(6), 2517–2542.

Lu, H.D., Chen, G., Tanigawa, H., and Roe, A.W. (2010). A motion direction map in macaque V2. Neuron 68, 1002–1013.

M H Munk, L G Nowak, P Girard, N Chounlamountri, J Bullier Visual latencies in cytochrome oxidase bands of macaque area V2. Proc Natl Acad Sci U S A. 1995 92: 988-992.

Marcar, V.L., Raiguel, S.E., Xiao, D., and Orban, G.A. (2000). Processing of kinetically defined boundaries in areas V1 and V2 of the macaque monkey. J. Neurophysiol. 84, 2786–2798.

Peterhans, E., Heider, B., and Baumann, R. (2005). Neurons in monkey visual cortex detect lines defined by coherent motion of dots. European Journal of Neuroscience, 21(4), 1091–1100.

Peterhans, E., and von der Heydt, R. (1993). Functional Organization of Area V2 in the Alert Macaque. European Journal of Neuroscience, 5(5), 509–524.

Roe, A. W., and Ts’o, D. Y. (2015). Specificity of V1-V2 orientation networks in the primate visual cortex. Cortex, 72, 168–178.

Shipp, S. and Zeki, S. (1985). Segregation of pathways leading from area V2 to areas V4 and V5 of macaque monkey visual cortex. Nature 315, 322–325.

Shipp S, Zeki S (1989) The organization of connections between areas V5 and V2 of macaque monkey visual cortex. Eur J Neurosci 1:333–354.

Shipp, S., and Zeki, S. (2002). The functional organization of area V2, II: The impact of stripes on visual topography. Visual Neuroscience, 19(2), 211–231.

Sincich LC, Horton JC. The circuitry of V1 and V2: integration of color, form, and motion. Annual Review of Neuroscience. 2005;28:303–326.

Thomas OM, Cumming BG, Parker AJ. A specialization for relative disparity in V2. Nat Neurosci 5: 472–478, 2002.

Von der Heydt, R., Peterhans, E., and Baumgartner, G. (1984). Illusory contours and cortical neuron responses. Science, 224(4654), 1260–1262.

Von der Heydt, R., and Peterhans, E. (1989). Mechanisms of Contour Perception Lines of Pattern Discontinuity. Journal of Neuroscience, 9(5), 1731-1748.

Von der Heydt R, Zhou H, Friedman HS (2000) Representation of stereoscopic edges in monkey visual cortex. Vision Res 40:1955–1967.

Zhou, H., Friedman, H. S., and von der Heydt, R. (2000). Coding of Border Ownership in Monkey Visual Cortex. The Journal of Neuroscience, 20(17), 6594–6611.

[Editors' note: further revisions were suggested prior to acceptance, as described below.]

We found that you have thoroughly revised the manuscript and performed additional analyses according to the Reviewer's comments, which is appreciated. The manuscript has been improved but there are some remaining issues that need to be addressed, as outlined below:1. We are somewhat concerned about the lack of significant difference in the receptive field (RF) size of V1 and V2 neurons. The reported 2.5 deg mean RF size for V1 cells is certainly inconsistent with what has been reported in the literature for parafoveal V1 neurons (more around 0.8-1 deg). This discrepancy to the published literature should be discussed in the paper – alongside the potential reasons for this, specifically:a. The most likely cause for this is eye movements (in fact the fixation window is reported to be 1.2deg, which is rather coarse for V1). The question here is in what way this imprecision in the mapping of V1 RFs may have affected the results, e.g. the reported differences between V1 and V2 cells. You should at the very least add a discussion of this issue.

We agree with the reviewers that the measured RF size for V1 (2.5 deg) was larger than those previously reported. Eye movements could be a cause, but it would equally increase the measurements of RF sizes for V2 neurons. In our recordings, the mean RF size for V2 neurons was 2.47 deg. Thus, we suspect that other factors (see below) might contribute more to the large V1 RF we measured. In the revised manuscript, whether the imprecision of V1 RF mapping affected the results was discussed (page 19 line 537-556).

b. The illustrated placement of the arrays looks like that for 3 out of 4 arrays very few electrodes would be safely in V1, which could also explain the lack of a difference in RF size. You should specify in the methods of the paper how exactly they determined functionally whether each electrode was in V1 or V2.

We added detailed description in Methods on how we determined that the electrodes were in V1 (page 34 line 1116-1123). We also added V1 ocular dominance maps into Figure 1—figure supplement 2 to show how the V1/V2 borders were determined.

Although these maps show clear V1/V2 borders, the actual transition from V1 neurons to V2 neurons was unlikely that sharp, especially for neurons in superficial layers. There should be a narrow “transition zone” in which V1 and V2 neurons were actually mixed or inseparable. Plus, there were always precision limits in our map-array alignments. Thus, it is possible that some of our V1 neurons were actually V2 neurons, or the other way around. We analyzed a subset of V1 cells (n=39) that were recorded from the electrodes further away from the V1/V2 border and found that their mean RF size was relatively smaller (2.05 deg).

The stimulus we used in measuring the RF size was a 0.8 deg grating patch, its size was also a little too large for V1 RF mapping. We added above information into discussion (page 19 line 537-556).

c. The V1 RF measurements shown in the authors' response in comparison with V2 should also be included in the supplementary figure (could be added to the supplement to Figure 3).

We have added the RF measurements into Figure 3—figure supplement 1.

2. Authors' Reply point 7). Here, you acknowledge that the zero lag in the CCG between DS and MB neuron responses is inconsistent with your hypothesis of DS inputs contributing to the MB responses. Yet, in the Results section of the paper you do not say so, rather you still maintain that the results are consistent with your hypothesis but concede that alternative/additional mechanisms may also be at play. If you indeed agree this result is inconsistent with your hypothesis, you should state so. Alternatively, you should clarify in the manuscript why you think this result may still be consistent with your hypothesis.

We revised the Results section to clearly indicate that the zero time lag is inconsistent with a simple DS-MB contribution model (page 15 line 407-410).

3. Authors' Reply point 8. The two alternative model schemes do a good job at clarifying why and you think that surround suppression may play a role in the generation of MB responses and how. We recommend to include this figure in the discussion of the paper.

We included this figure as Figure 5—figure supplement 4 and linked it to the relevant discussion (page 21 line 615-630).

4. Authors' Reply point 11. Please include in the manuscript your explanation provided here of why the 4 neurons in group E were included in the MB population analysis. This is ok, but this needs to be clear in the manuscript (methods and figure legend need to be consistent).

We have added the explanation into the Method section and the corresponding figure legend.

5. line 1041-1051: ".… sorted response clusters were usually multi-unit responses,.…" "Neurons with inter-spike-intervals larger than 2.5 ms were identified as SUs." The paragraph reads as if most data presented in the paper were multi-unit, but were re-classified as SU based on spike timing only, which by itself is not enough. The main results (line 106) however state that SUs were identified on unique spike wave form and asserts that most cells are SU. We presume this means that the SUs were clearly separable from the remaining MU or noise.There need to be clear, consistent statements in both Results and Methods, as the reader needs to know (i) whether they are dealing with SU or MU results and (ii) if both were combined whether specific results about basic MB tuning differ for clear SUs und MUs.

We thank the reviewer for pointing out these unclear statements. In the revised manuscript, we have clarified the definition of SU (page 33 line 1073-1079), how much percentages of SUs in the 3 datasets, and whether inclusion of MUs affects the results (page 5 line 100-119 and page 33-34 line 1087-1100).

6. The reviewers would be grateful if you could please always point to the pages in the revised manuscript where the revisions were made. It really saves reviewers a lot of time.

We are very sorry for the inconvenience we made, we did so in this response letter.